# The long-term effects of chemotherapy on normal blood cells

Emily Mitchell[1,2,3,14], My H. Pham [1,14], Anna Clay[2,4,14], Rashesh Sanghvi [1], Nicholas Williams [1], Sandra Pietsch[2,3], Joanne I. Hsu[5], Nina Friesgaard Øbro[2,6], Hyunchul Jung[1], Aditi Vedi [2,7], Sarah Moody [1], Jingwei Wang [1], Daniel Leonganmornlert [1], Michael Spencer Chapman [1], Ellie Dunstone[1], Anna Santarsieri [2,3], Alex Cagan [1], Heather E. Machado[1], E. Joanna Baxter[3], George Follows[3], Daniel J. Hodson [2,3], Ultan McDermott [1,8,9], Gary J. Doherty [8], Inigo Martincorena [1], Laura Humphreys[1], Krishnaa Mahbubani [10,11], Kourosh Saeb Parsy [10,11], Koichi Takahashi [12], Margaret A. Goodell [5], David Kent [13], Elisa Laurenti [2,3], Peter J. Campbell [1], Raheleh Rahbari [1], Jyoti Nangalia [1,2,3,14] ✉ & Michael R. Stratton [1,14] ✉

Several chemotherapeutic agents act by increasing DNA damage in cancer cells, triggering cell death. However, there is limited understanding of the extent and long-term consequences of collateral DNA damage in normal tissues. To investigate the impact of chemotherapy on mutation burdens and the cell population structure of normal tissue, we sequenced blood cell genomes from 23 individuals aged 3–80 years who were treated with a range of chemotherapy regimens. Substantial additional somatic mutation loads with characteristic mutational signatures were imposed by some chemotherapeutic agents, but the effects were dependent on the drug and blood cell types. Chemotherapy induced premature changes in the cell population structure of normal blood, similar to those caused by normal aging. The results show the long-term biological consequences of cytotoxic agents to which a substantial fraction of the population is exposed as part of disease management, raising mechanistic questions and highlighting opportunities for the mitigation of adverse effects.

Over the course of a lifetime, one in two people develops cancer. A long-standing approach to cancer treatment is systemic administration of a diverse group of cytotoxic chemicals, often termed 'chemotherapy', which includes alkylating agents, platinum compounds, antimetabolites, topoisomerase inhibitors, vinca alkaloids and cytotoxic antibiotics[1]. Some of these agents exert their therapeutic effects by causing damage to DNA that, in turn, triggers the death of malignant cells[2]. Approximately 30% of individuals with cancer, and thus approximately 10% of the whole population in developed countries, are exposed to chemotherapy at some point in their lifetime (www.cancerresearchuk.org/health-professional/cancer-statistics-for-the-uk), providing considerable exposure of normal tissues to the actions of these drugs.

Chemotherapy can have long-term side effects on normal tissues. It confers an increased risk of cancers of the blood[3–6], lung, bladder and colon[7,8] and is sometimes toxic to the kidney, blood, heart, brain, gastrointestinal tract, peripheral nervous system and gonads, engendering long-term deterioration in organ function[9–14]. There is limited understanding of the biological mechanisms underlying these sequelae. It is plausible that some are related to the consequences of DNA damage and thus could be elucidated through the genome sequences of normal tissues, which may reveal changes in somatic mutation burdens or clonal composition immediately or decades following chemotherapy.

Sequencing of cancers arising after chemotherapy treatment has revealed variably elevated somatic mutation loads, in some instances

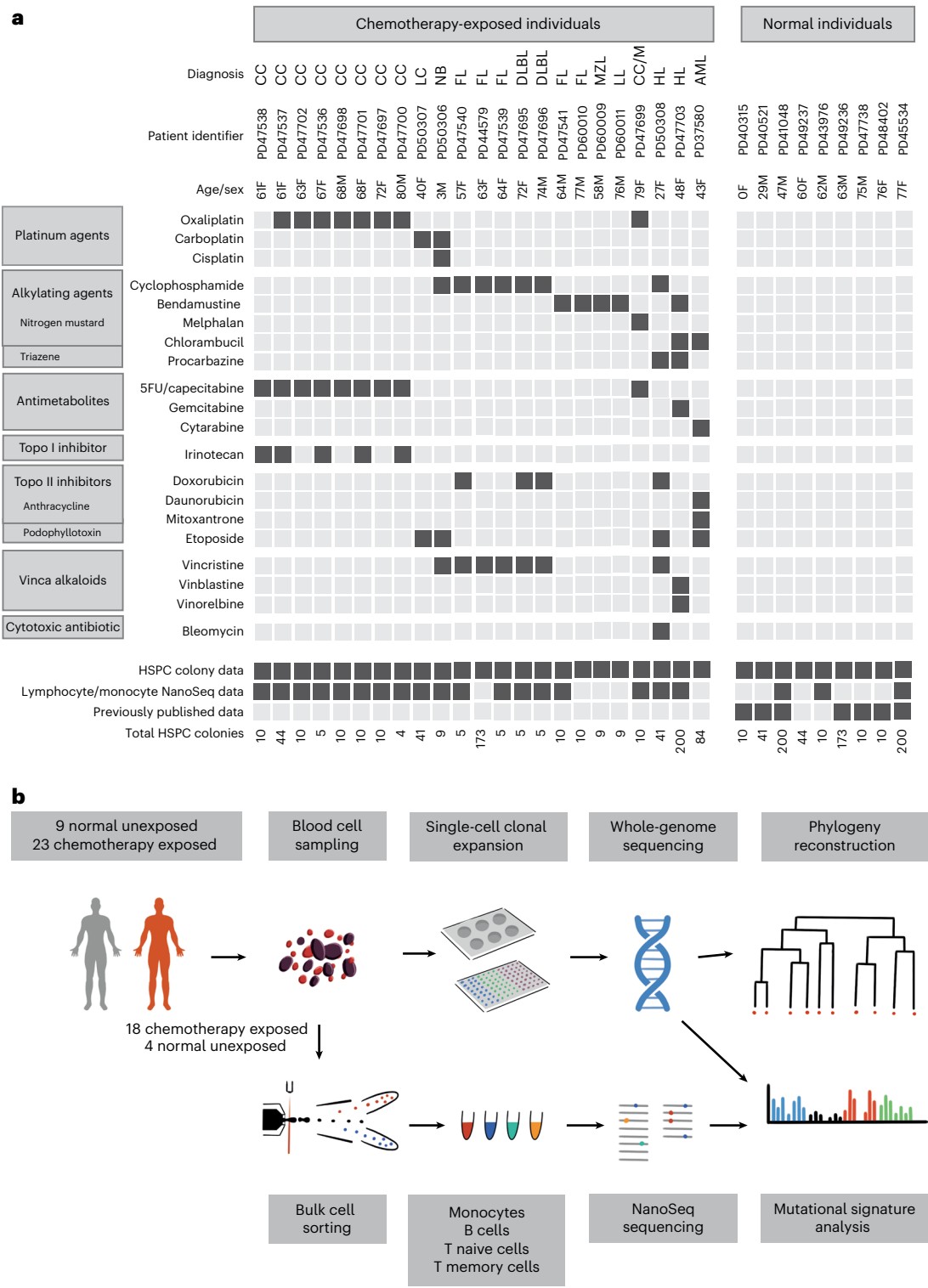

**Fig. 1 | Donor information and experimental approach. a**, Donor demographic details, chemotherapy exposure and sample information. CC, colorectal carcinoma; LC, lung cancer; NB, neuroblastoma; FL, follicular lymphoma; DLBL, diffuse large B cell lymphoma; MZL, marginal zone lymphoma; LL, lymphoplasmacytic lymphoma; M, multiple myeloma; HL, Hodgkin lymphoma; 5FU, 5-fluorouracil; Topo, topoisomerase. **b**, Experimental approach.

characterized by distinct mutational signatures[15–18]. However, there is little direct information concerning the mutagenic effects of chemotherapy on normal tissues in vivo. Studies of a small number of individuals show that normal colorectal epithelium, blood and sperm can exhibit additional somatic mutations after chemotherapy[19,20]. Furthermore, chemotherapy can alter the clonal structure of normal cell populations, as illustrated in the blood, where treatment increases the incidence of clonal hematopoiesis, favoring clones with driver mutations in the DNA damage response genes *PPM1D*, *TP53* and *CHEK2* (refs. 21–23).

To survey the long-term impacts of chemotherapeutic agents on normal body tissues, we here investigate their effects on normal blood by whole-genome sequencing (WGS) of cells from chemotherapy-exposed individuals. The blood offers several desirable

features in this regard, including ease of randomly sampling cells from the whole tissue, predictable mutation accumulation in unexposed individuals[24], opportunities to interrogate different cell subtypes and maturation states, and feasibility of surveying changes in cell population clonal structure.

## Results

### Genome sequencing of chemotherapy-exposed blood

To conduct a primary survey of chemotherapy effects on normal blood cell genomes, we analyzed 23 individuals with hematological or solid malignancies who had collectively been exposed to multiple chemotherapy classes and multiple members of each class, with variable time intervals since exposure. These individuals were aged 3–80 years and had been treated with commonly used chemotherapy regimens for hematological malignancies (Hodgkin lymphoma, $n = 2$; follicular lymphoma, $n = 5$; diffuse large B cell lymphoma, $n = 2$; lymphoplasmacytic lymphoma, $n = 1$; marginal zone lymphoma, $n = 1$; multiple myeloma, $n = 1$; acute myeloid leukemia (AML), $n = 1$) and solid cancers (colorectal carcinoma, $n = 9$; neuroblastoma, $n = 1$; lung cancer, $n = 1$). One individual had been treated with chemotherapy for both multiple myeloma and colorectal carcinoma. The individual with AML had also been treated with chemotherapy for Behcet disease, a noncancer condition (Fig. 1a and Supplementary Table 1). Most had received a combination of agents and, collectively, had been exposed to 21 drugs from all of the main chemotherapy classes, including alkylating agents (cyclophosphamide, $n = 8$; chlorambucil, $n = 2$; bendamustine, $n = 5$; procarbazine, $n = 2$; melphalan, $n = 1$), platinum agents (oxaliplatin, $n = 7$; carboplatin, $n = 2$; cisplatin, $n = 1$), antimetabolites (capecitabine, $n = 7$; 5-fluorouracil, $n = 6$; gemcitabine, $n = 1$; cytarabine, $n = 1$), topoisomerase I inhibitors (irinotecan, $n = 5$), topoisomerase II inhibitors (etoposide, $n = 4$; doxorubicin, $n = 4$; daunorubicin, $n = 1$; mitoxantrone, $n = 1$), vinca alkaloids (vincristine, $n = 7$; vinblastine, $n = 1$; vinorelbine, $n = 1$) and cytotoxic antibiotics (bleomycin, $n = 1$). The time intervals from chemotherapy exposure to tissue sampling ranged from less than 1 month to 6 years for most cases. However, one individual sampled at age 48 years had been treated for Hodgkin lymphoma at ages 10 and 47 years. Additionally, the individual sampled at age 43 years following induction chemotherapy for AML had also received long-term chlorambucil for Behcet disease diagnosed at age 13 years. Seven patients had also received localized radiotherapy (Supplementary Table 1). We endeavored to exclude a chemotherapy agent being administered in the context of only a single cancer type to avoid any confounding effects, but this was not always possible (Fig. 1a). Results were compared to those from nine healthy, non-chemotherapy-exposed individuals (Fig. 1a and Supplementary Table 1).

Three experimental designs for detecting and analyzing somatic mutations were used. First, 189 single-cell-derived hematopoietic stem and progenitor cell (HSPC) colonies from the 23 chemotherapy-exposed individuals and 90 colonies from the 9 controls were expanded and individually subjected to WGS at 23-fold average coverage to compare mutation burdens and mutational signatures (Extended Data Fig. 1a,b). Second, from six individuals exposed to a range of chemotherapeutic agents, a further 589 single-cell colonies

underwent WGS (41–259 colonies per individual; mean sequencing depth 15-fold). These phylogenies were compared to similar-sized phylogenies (608 colonies) from five normal individuals across a similar age range to survey the effect of chemotherapy on the clonal structure of the HSPC population. Third, flow-sorted subpopulations of B cells, T memory cells, T naive cells and monocytes from whole-blood samples from 18 chemotherapy-exposed individuals and 3 unexposed normal individuals (Fig. 1b) underwent WGS using duplex sequencing, which allows reliable identification of somatic mutations in polyclonal cell populations[25].

### Chemotherapy-induced somatic mutations in the blood

Somatic single-base substitution (SBS) mutations in HSPCs from normal adults accrue constantly at a rate of ~18 per year, leading to a burden of ~1,500 SBSs in 80-year-old individuals[24]. HSPCs from 17 of the 23 chemotherapy-exposed individuals showed elevated mutation burdens compared to those expected for their ages ($P < 2.2 \times 10^{-16}$, mixed-effects model) (Fig. 2). Four showed large increases of >1,000 SBSs (Fig. 2a), thirteen showed more modest increases of 200–600 SBSs (Fig. 2b), and six showed no increases (Extended Data Fig. 1c–f). The burdens of small indels in HSPCs were also increased in the four individuals with the greatest elevations in SBS burdens (Extended Data Fig. 2a,b). Increases in structural variant and copy number changes were not observed, including in those individuals exposed to topoisomerase II inhibitors, which have been implicated in the development of secondary malignancies driven by specific oncogenic rearrangements[26] (Extended Data Fig. 2c,d). However, the small number of individuals may have limited the statistical power to identify minor differences.

Nineteen of the twenty-three chemotherapy-treated individuals received multiple agents. Therefore, in many cases, it was uncertain which agents were responsible for the elevated mutation loads. To address this, we extracted mutational signatures from the SBS and indel mutation catalogs of chemotherapy-exposed individuals and controls and estimated the contribution of each signature to the somatic mutations in the blood cells of each individual (Fig. 2c and Extended Data Fig. 3). We used prior knowledge of previously described mutational signatures attributed to normal endogenous mutational processes and to some mutagenic exposures[27], as well as the specific chemotherapy regimens received by each individual, to associate each signature with its putative causative agent.

Twelve SBS mutational signatures were extracted (Fig. 2c and Supplementary Table 2). Four were composed of known signatures of normal HSPCs and mature lymphocytes (Supplementary Table 3). The first was predominantly constituted by SBS1, characterized by C>T mutations at CG dinucleotides, together with a contribution from SBS5, which is relatively flat and featureless. SBS1 and SBS5 are found in most normal cell types thus far studied. The second was SBSBlood, a blood-specific signature predominant in HSPCs[28,29]. The third was SBS7a, an ultraviolet light-caused signature found in memory T cells that have presumably resided in the skin during life[30]. The fourth was SBS9, a signature of somatic hypermutation found in B cells (Fig. 2c). Three indel mutational signatures were extracted (Supplementary

**Fig. 2 | Mutation burden and mutational signatures in normal and chemotherapy-exposed blood cells. a**, Burden of SBS across normal individuals and the four chemotherapy-exposed individuals with the highest SBS burdens. The points represent individual HSPC colonies. The boxes indicate the median and interquartile range; the whiskers denote the minimum and maximum values. The black line represents a regression of age on mutation burden across the unexposed individuals, with the 95% confidence interval shaded. Annotations indicate the corresponding individuals from Fig. 1, providing details on the type of malignancy (as previously defined) and chemotherapy treatment (1, platinum agents; 2, alkylating agents; 3, antimetabolites; 4, topoisomerase I inhibitors; 5, topoisomerase II inhibitors; 6, vinca alkaloids; 7, cytotoxic

antibiotics). **b**, Depiction of data as in **a**, but the y axis is cut off at 2,000 SBSs for better visualization of the majority of the chemotherapy-exposed cohort data. The points represent individual HSPC colonies. The boxes indicate the median and interquartile range; the whiskers denote the minimum and maximum values. The gray shading in **b** represents the 95% CI of the regression of age on mutation burden across the unexposed individuals. The black line represents a regression of age on mutation burden, with the 95% confidence interval shaded. **c**, Mutational signatures extracted using the hierarchical Dirichlet process (HDP) from the full dataset of normal and chemotherapy-exposed HSPC colonies and duplex sequencing of bulk mature blood cell subsets.

Table 4). Two were similar to known indel signatures and were present in both normal and chemotherapy-exposed individuals: the first comprised ID1, and the second was a composite of ID3, ID5 and ID9 (ID3/5/9; Extended Data Fig. 3).

Eight SBS mutational signatures were interpreted as being present exclusively in chemotherapy-treated individuals (Fig. 2c), based on the observation that they accounted for <1% of mutations in HSPCs from adult controls (Extended Data Fig. 4). Four of these have not been

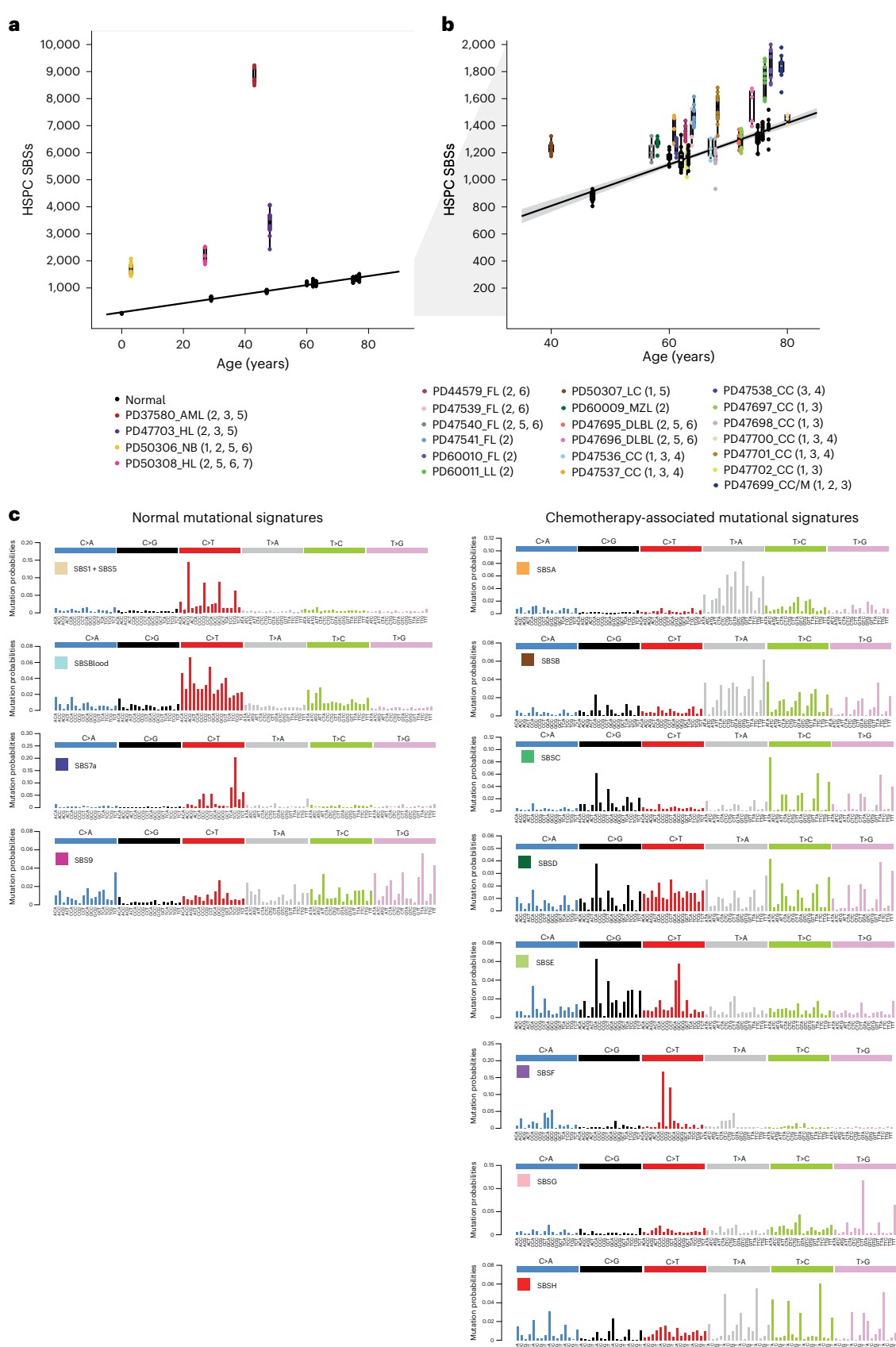

reported previously and are not represented in the COSMIC SBS mutational signature database. SBSA is likely due to the triazene alkylating agent procarbazine. There were three similar but distinct signatures relating predominantly to specific nitrogen mustard alkylating agents: SBSC to chlorambucil, SBSD to bendamustine and SBSE to melphalan. SBSF is associated with the platinum agents cisplatin and carboplatin, and SBSG is associated with the antimetabolite 5-fluorouracil or its prodrug capecitabine. The etiologies of SBSB and SBSH are less clear-cut and are discussed further below. Excess SBSs and specific SBS mutational signatures were not obviously associated with topoisomerase inhibitors (which cause DNA strand breaks), vinca alkaloids (which inhibit microtubule formation during cell division) and the cytotoxic antibiotic bleomycin (which is thought to bind and cleave DNA). Only one high-confidence indel mutational signature was found exclusively in chemotherapy-treated individuals: IDA, associated with procarbazine exposure.

SBSA contributed substantial additional mutation loads to blood cells from two individuals treated for Hodgkin lymphoma (PD50308 and PD47703) (Fig. 3). The only chemotherapy common to their treatment regimens was the alkylating agent procarbazine; no other individuals had been treated with procarbazine, and HSPC phylogenies indicated that SBSA mutations occurred early during PD47703's life, consistent with procarbazine treatment at age 10 years (Extended Data Fig. 5a). The observed mutational signatures were compared to those previously reported, using a combination of visual inspection, data review and the cosine similarity between trinucleotide mutation profiles. Cosine similarities >0.90 between two mutational signatures are highly unlikely to occur by chance, and a cosine similarity of >0.95 generally suggests the same underlying process. SBSA exhibits similarity to COSMIC signature SBS25 (cosine similarity 0.84), which has previously been associated with procarbazine[19,31]. An indel signature (IDA) was also identified as being most likely attributable to procarbazine, being found only in the two individuals treated with procarbazine (Extended Data Fig. 3). Alkylating agents cause alkyl DNA adducts, resulting in base mispairing and DNA breaks. Procarbazine is a triazene/hydrazine monofunctional alkylating agent.

SBSB was found predominantly in the individual exposed to chlorambucil, procarbazine and bendamustine (PD47703). SBSB, like SBSA (procarbazine), is predominantly composed of T>A substitutions, with a cosine similarity to SBSA of 0.82 and cosine similarities to SBSD (bendamustine; 0.82) and SBSC (chlorambucil; 0.74) suggesting that it is unlikely to be due to any of these in isolation. It is also present at low levels in the T memory cells of the other procarbazine-exposed individual who was also exposed to cyclophosphamide (PD50308). It seems plausible that SBSB may result from an interaction between two classes of alkylating agents.

Of the nitrogen mustard-associated signatures, SBSC contributed all mutations to the individual who received chlorambucil from childhood (PD37580); SBSD contributed all excess mutations to one of the individuals exposed to only bendamustine (PD60010) and was also present at a much lower burden in a subset of cyclophosphamide-exposed individuals; SBSE was found only in the single individual exposed to low-dose melphalan (PD47699). Nitrogen mustard alkylating agents have two reactive sites and are, in consequence, bifunctional, forming intrastrand and interstrand DNA cross-links in addition to simple adducts. The SBSC and SBSD signatures identified here are similar to a recently published mutational signature found in the germlines of two individuals whose fathers had been treated with two different nitrogen mustard agents (chlorambucil and ifosfamide)[20,32], and SBSE is similar (cosine similarity 0.84) to the previously described signature in multiple myeloma genomes with prior melphalan exposure[33–35].

SBSF was found in individuals treated with carboplatin or cisplatin and in a subset of oxaliplatin-treated individuals in whom it was present at much lower burdens. It is highly similar to COSMIC SBS31 (cosine similarity 0.95), which has previously been associated with prior platinum

exposure in cancer genomes[32,36] (Fig. 3). Platinum compounds act by binding DNA and forming intrastrand and interstrand DNA cross-links, in a similar manner to bifunctional alkylating agents. However, SBSF/SBS31 is different from the bifunctional nitrogen mustard signatures, indicating that the patterns of DNA damage and/or DNA repair induced by platinum agents and nitrogen mustards differ.

SBSG is highly similar to COSMIC SBS17 (cosine similarity 0.93), which has previously been found in the genomes of cancers exposed to 5-fluorouracil[37] and in the normal intestine of one 5-fluorouracil-exposed individual[19]. It was undetectable in HSPCs and found at the highest burdens in lymphoid cells from individuals treated with 5-fluorouracil or its prodrug capecitabine (Fig. 4). 5-Fluorouracil is a pyrimidine analog misincorporated into DNA in place of thymine, consistent with causing a mutational signature characterized predominantly by thymine mutations.

SBSH was detectable only in the T cells of a single individual who was also the only person to have received gemcitabine, a cytosine analog. However, the origin of SBSH remains uncertain.

The isolation of multiple HSPC colonies from each individual allowed for assessing the variation in mutagenic exposures across each of their HSPC populations. Although there was some variability in the mutation burdens attributable to cisplatin/carboplatin, procarbazine, chlorambucil and bendamustine (the most highly mutagenic agents) across HSPCs from each individual (Fig. 3), the evidence suggests that there were no HSPCs completely protected from DNA damage. The multiple HSPCs from each individual also allowed the formation of phylogenetic trees, permitting the timing of mutagenic impacts. The phylogenetic timings were in keeping with the known periods of exposure: PD47703 with both early-life exposure to procarbazine and chlorambucil and later-life exposure to bendamustine, PD37580 with both early- and late-life exposure to chlorambucil, and PD47699 with late-life exposure to melphalan (Fig. 3).

Although limited numbers of individuals, different drug combinations and different dose regimens preclude definitive evaluation, the inclusion of individuals treated with different members of the same chemotherapy class enabled a preliminary comparison of their effects. Among the nitrogen mustard alkylating agents, chlorambucil, bendamustine and melphalan caused substantially greater alkylating agent-associated mutation burdens in normal blood cells than cyclophosphamide, which engendered only minimal (<5% and hence not shown in the figures) increases in mutation load (Fig. 3, Extended Data Fig. 5b and Supplementary Tables 5 and 6). Similarly, carboplatin and cisplatin caused much higher SBSF mutation burdens than oxaliplatin, which conferred SBSF mutation burdens of <5% in all cases, despite prolonged oxaliplatin treatment (up to 22 cycles) in some individuals (Fig. 3, Extended Data Fig. 6 and Supplementary Tables 5 and 6). Therefore, chemotherapeutic agents of the same class, some used interchangeably in cancer treatment, may confer substantially different mutation burdens in normal blood cells.

Flow sorting of monocytes, B cells, T memory cells and T naive cells enabled us to investigate the responses of different cell types to identical chemotherapy exposures. Overall, the patterns of SBS signature burdens in monocytes were similar to those in HSPCs, whereas the patterns in B and T lymphocytes showed differences for some agents (Supplementary Table 7). For example, SBSG, caused by 5-fluorouracil/capecitabine, contributed additional mutation burdens in B lymphocytes (P = 0.0), T naive cells (P = 0.0097) and T memory lymphocytes (P = 0.0014), but was undetectable in HSPCs and monocytes (Fig. 4). In contrast, SBSF, caused by the platinum agents, contributed larger mutation burdens in HSPCs, monocytes and B cells than in T naive and T memory cells, although we only have T cell data for one carboplatin-exposed individual (Fig. 4). The mutation loads contributed by SBSA, caused by procarbazine, were similar across cell types. Therefore, some chemotherapeutic agents engender different mutation burdens in different cell types.

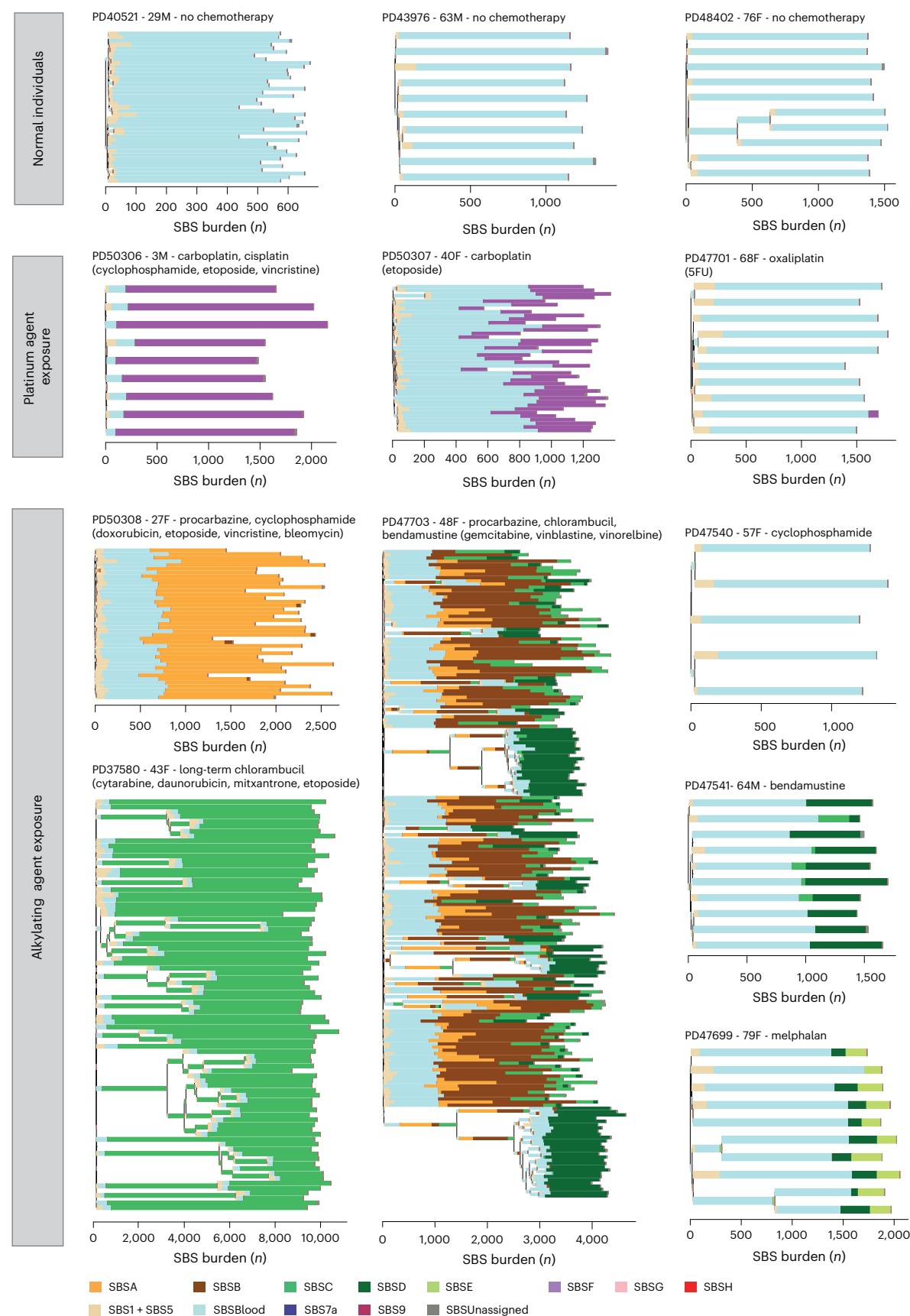

**Fig. 3 | Phylogenetic trees and mutational signatures across a range of normal and chemotherapy-exposed individuals.** Phylogenies were constructed using shared mutation data and the algorithm MPBoot (Methods). Branch lengths correspond to SBS burdens (*x* axes). A stacked bar plot represents the signatures contributing to each branch, with the color code below the trees. SBSUnassigned indicates mutations that could not confidently be assigned to any reported signature. Drugs in parentheses are those received by the individual at the same time but not believed to be the mutagenic agents.

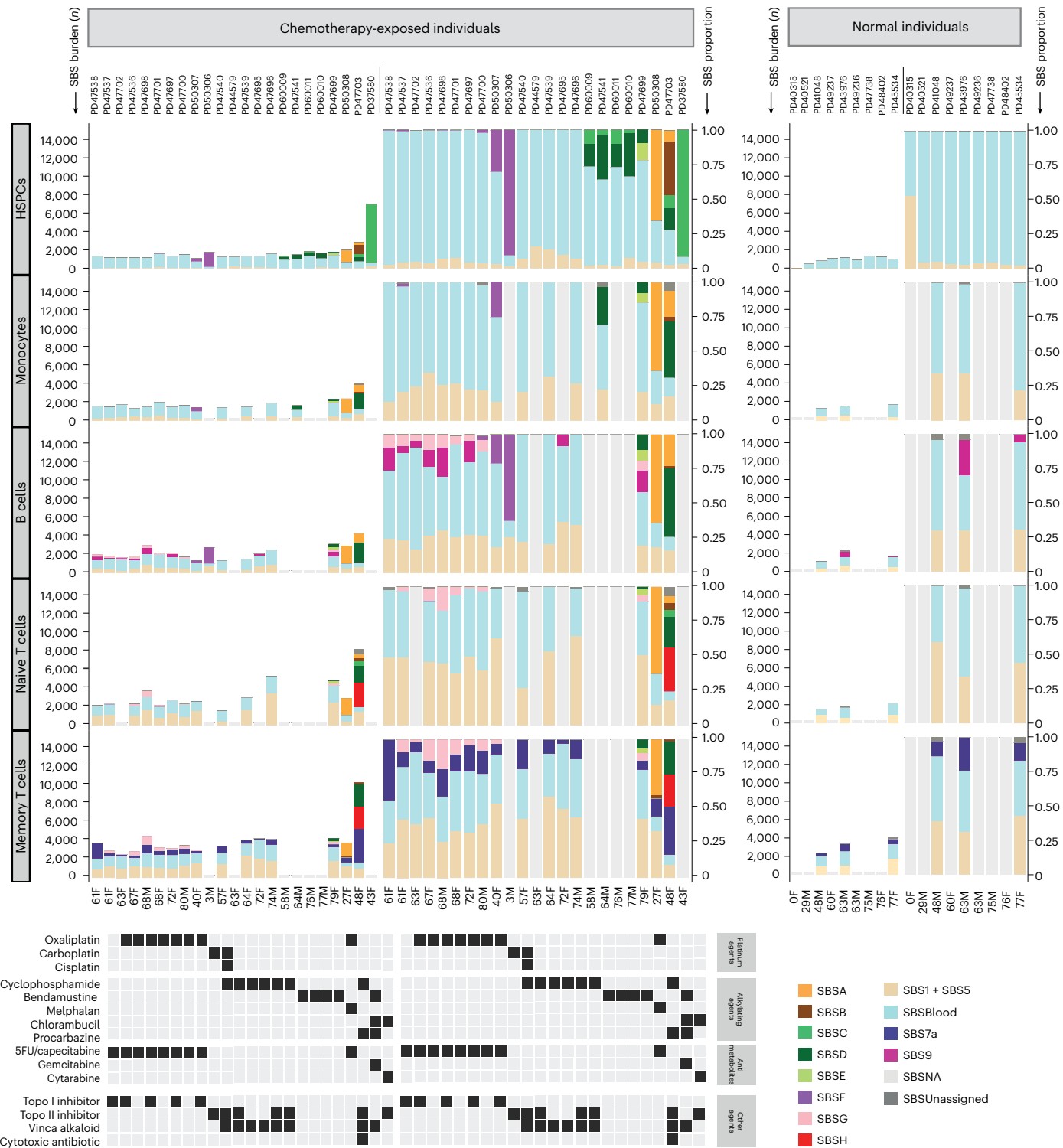

**Fig. 4 | Mutation burden and SBS mutational signatures across different blood cell types.** Stacked bar plots represent the absolute contributions of each SBS mutational signature to the SBS mutation burden across cell types (left), compared to the proportionate contribution of each signature (right). HSPC data were generated by pooling HSPC WGS colony data from each individual. Mature blood cell data were generated using duplex sequencing of ~40,000 cells of each type. For the normal unexposed individuals, the T cell subset data are from CD4[+] T cells; for the chemotherapy-exposed individuals, the T cell subsets contain both CD4[+] and CD8[+] T cells. SBSUnassigned indicates mutations that could not confidently be assigned to any reported signature. SBSNA indicates that duplex sequencing data are unavailable for this subset. In seven individuals, granulocyte mutation profiles were available, which were not discernibly different from the mutational spectra observed in HSPCs and monocytes from those individuals. Due to the lack of availability of this cell type for most patients, the data are not shown.

## Hematopoietic clonal architecture after chemotherapy

To investigate the effect of chemotherapies on the architecture of cell populations, we generated extensive phylogenies of HSPCs from six chemotherapy-exposed individuals and compared them to the HSPC phylogenies of nonexposed individuals of similar ages. An exemplar HSPC phylogeny of a normal, non-chemotherapy-exposed 48-year-old individual showed only one barely detectable clonal expansion and no 'driver' mutations in cancer genes (Fig. 5a). Such trees are typical of healthy middle-aged adults[24].

Given that changes in the clonal composition of the HSPC population due to chemotherapy-induced bottlenecks and positive selection may take many years to become apparent, we focused on two individuals sampled 30 and 39 years after their earliest exposure to chemotherapy (PD37580 and PD47703; Fig. 5c,d). In contrast to normal middle-aged individuals, a 48-year-old woman (PD47703) treated for Hodgkin lymphoma with chlorambucil and procarbazine at age 10 years and bendamustine at age 47 years showed multiple independent clonal expansions carrying 'driver' mutations in the DNA damage response gene *PPM1D* (Fig. 5c). A similar pattern, with expanded *PPM1D* and *TP53* mutant clones, was observed in a 43-year-old woman (PD37580) after long-term chlorambucil treatment (Fig. 5d). This pattern of multiple, large clonal expansions is characteristic of normal individuals aged >70 years[24]. However, in healthy older adults, clonal expansions exhibit predominantly *DNMT3A* and *TET2* driver mutations or no apparent driver (Fig. 5b). While *TP53* and *PPM1D* mutations can also be observed in the context of normal aging, the pattern of dozens of parallel clonal expansions harboring these mutations is likely to reflect the unique selective landscape induced by some chemotherapies.

Chemotherapy could induce this prematurely aged HSPC cell population profile by increasing mutation loads and/or by altering microenvironmental selection. Chemotherapy favors the survival of clonal hematopoiesis of indeterminate potential (CHIP) clones with driver mutations in *PPM1D*, *TP53* and *CHEK2* (ref. 21), which usually predate the chemotherapy. Similarly, the HSPC phylogenetic tree of PD37580 indicates that at least two *PPM1D* driver mutations arose before the chemotherapy given during childhood (Extended Data Fig. 7a). Furthermore, in PD47703, a comparison of two samples taken 1 year apart, during which additional chemotherapy (cyclophosphamide, doxorubicin and vincristine) had been administered, revealed an approximately 50% increase in the size of preexisting *PPM1D* mutated clones and no new mutant clones (Extended Data Fig. 7b). Thus, chemotherapy-induced changes in selection appear more influential than chemotherapy-induced creation of new driver mutations in generating the prematurely aged HSPC profile.

The prematurely aged architecture of the HSPC population was not observed in two young adults (PD50308 aged 29 years and PD50307 aged 40 years) who received chemotherapy that caused substantial increases in mutation loads and was administered 2 years or less before sampling (Extended Data Fig. 8). It was also not observed in two further individuals who were treated with cyclophosphamide and oxaliplatin (PD44579 aged 63 years and PD47537 aged 61 years) and exhibited minimally increased mutation loads (Extended Data Fig. 9). Therefore, it is conceivable that multiple and/or prolonged chemotherapeutic exposures are required to generate the prematurely aged architecture. However, it is also possible that chemotherapy-engendered clonal expansions require decades to become detectable, as already demonstrated for clones under positive selection during normal aging[24,38].

Changes in clonal architecture resulting from chemotherapy exposure are relevant for two reasons: first, *PPM1D* mutant clones may themselves reduce the regenerative ability of the bone marrow[21,39], or the presence of *PPM1D* clones may simply be a marker of a more general state of reduced hematopoietic stem cell (HSC) function after chemotherapy. One may speculate that the presence of many such clones had a role in the development of cytopenias and infections in PD47703 following autograft treatment. Second, the selection of *TP53* mutant clones confers a high risk of developing secondary myeloid malignancies, including AML as seen in PD37580, whose disease was treatment-refractory and carried biallelic *TP53* mutations and a complex karyotype.

## Discussion

This initial survey demonstrates that some commonly used chemotherapies, at dose regimens used in clinical practice, increase somatic mutation burdens and alter the population structure of normal blood cells. Individuals with elevated mutation burdens have likely experienced very high mutation rates over short periods. For example, an additional 1,000 SBSs acquired in an HSPC due to chemotherapy administered during the course of 1 year is equivalent to an approximately 50-fold increase in the average mutation rate over the year, and it is plausible that mutation rates within hours or days of chemotherapy are even higher. The additional long-term mutation loads are also sometimes considerable. A 3-year-old boy treated for neuroblastoma had more than tenfold the number of somatic SBSs expected for his age, exceeding the burden in normal 80-year-old individuals.

The additional mutation burdens differed substantially both between chemotherapy classes and between agents of the same class. As an important mechanism underlying the therapeutic effect of many chemotherapies is thought to be DNA damage induction, it is notable that different agents of the same class, at their therapeutic doses, have such different impacts on mutation generation in normal cells. The reasons for this are unclear but may reflect subtle differences between agents in the nature of the DNA damage caused, the repairability of the damage, the extent of induction of normal cell death and the levels of normal cell exposure. For example, cyclophosphamide is thought to relatively spare HSPCs due to their higher levels of aldehyde dehydrogenase, an enzyme that inactivates a cyclophosphamide intermediary[40]. However, it may also be the case that the extent of DNA damage does not directly correlate with the level of cytotoxicity of some chemotherapies.

The additional mutation burdens caused by chemotherapies are characterized by distinct mutational signatures, often shared by agents of the same class. The signatures are similar to those induced by the same agents in cancer cells, suggesting that the patterns of induced DNA damage, and its processing into mutations through DNA repair and replication, are similar in normal and cancer cells, even if the tolerance of DNA damage by normal and cancer cells differs.

Increases in mutation loads imposed by chemotherapies differed between blood cell types, and the profile of differences between cell types differed between chemotherapeutic agents. The mechanisms underlying these complex landscapes are uncertain but may reflect intrinsic differences in the metabolic capabilities, DNA repair capacities and cell division rates of the different cell types.

Changes in hematopoietic clonal architecture characterized by increasing dominance of large clones, often with driver mutations in cancer genes, are a feature of normal aging. Chemotherapy caused a similar pattern of change prematurely, although with a different repertoire of mutated genes. However, these changes were not observed in all individuals. Whether these chemotherapy-induced changes in population architecture are contingent on the long duration and/or multiplicity of treatment, or simply occur with the passage of decades after treatment (which may allow clones with limited growth advantage under normal conditions to become detectable), requires further investigation.

This study has limitations. In addition to the small number of individuals, solid tumors were predominantly colorectal carcinoma, hematological cancers were predominantly of lymphoid origin, and not all mutagenic chemotherapies may have been included. We were unable to address the role of multiple other plausible factors influencing mutation burdens and selective effects, such as the pharmacokinetics of drug administration. We acknowledge that the in vitro culture of

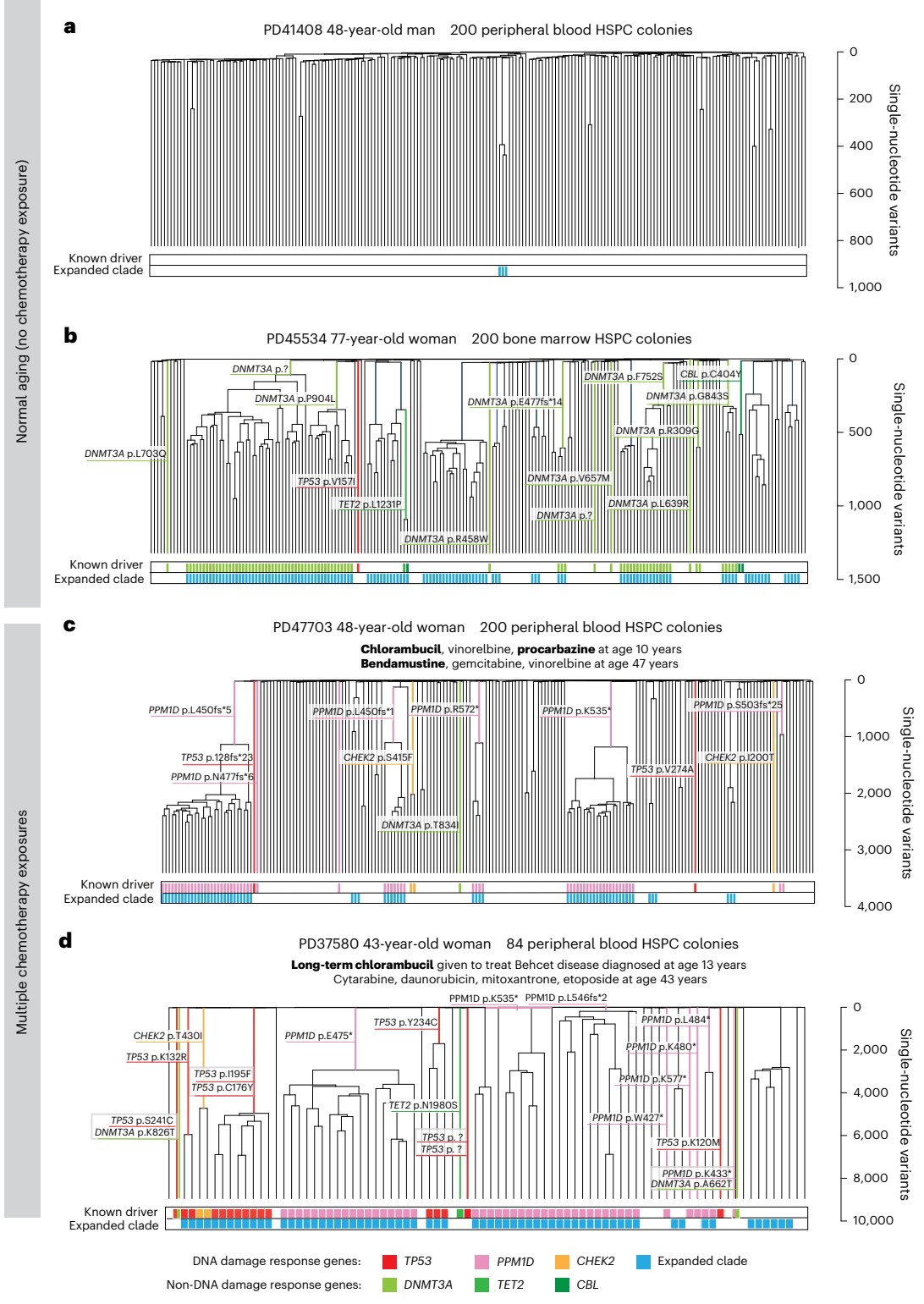

**Fig. 5 | HSPC phylogenies for two normal unexposed and two chemotherapy-exposed adult individuals. a,b,** Phylogenies for two normal unexposed donors: one young adult (**a**) and one older adult (**b**). **c,d,** Phylogenies for two young adult chemotherapy-treated individuals, both with more than one chemotherapy exposure. Phylogenies were constructed using shared mutation data and the algorithm MPBoot (Methods). Branch lengths reflect the number of mutations assigned to the branch, with the terminal branches adjusted for sequence coverage; the overall root-to-tip branch lengths have been normalized to the same total length (because all colonies were collected from a single time point). The *y* axis represents the number of SBSs accumulating over time. Each tip on a phylogeny represents a single colony, with the respective numbers of colonies of each cell and tissue type recorded at the top. Onto these trees, we have layered clone- and colony-specific phenotypic information. We have highlighted branches on which we have identified known oncogenic drivers in 1 of 18 clonal hematopoiesis genes (Supplementary Table 2) color-coded by gene. A heat map at the bottom of each phylogeny highlights colonies from known driver clades colored by gene and the expanded clades (defined as those with a clonal fraction of >1%) in blue. In the individual in **d**, the AML was derived from the biallelic *TP53*-mutated clade carrying *TP53* p.I195F and *TP53* p.C176Y. Drugs not highlighted in bold text are those received by the individual at the same time but not believed to be the mutagenic agents.

HSPCs may have introduced an unintended sampling bias, and future studies should address how the observed changes in the clonal composition of HSPCs following chemotherapy translate to differences in circulating mature blood cells.

Together with other work[17,32,41], our study points to a future agenda for systematic genomic analysis of normal tissues after chemotherapy. These studies could incorporate a wider range of at-risk normal tissues, sampled before and after treatment, after short and long periods, and across the full range of chemotherapies. Within the single-disease setting of clinical trials, this approach could aim to improve combination chemotherapeutic regimens and address dose–mutation relationships. Comprehensive prospective surveys of this nature would improve the understanding of the consequences of widespread, self-administered mutagenic exposure in human populations and provide a scientific basis for optimizing long-term patient health.

In conclusion, some chemotherapies impose additional mutational loads and change the cell population structure of normal blood. Both impacts plausibly contribute to long-term consequences, including second malignancies, infertility and loss of normal tissue resilience. Clinical data support this view, with the most mutagenic agents in this study having measurably greater long-term treatment toxicities. For example, of the bifunctional alkylating agents, melphalan and chlorambucil are associated with higher risks of secondary malignancies than cyclophosphamide[3,42,43]. In addition, procarbazine has been associated with a particularly high risk of second cancer and infertility and is, for this reason, no longer used in the treatment of pediatric Hodgkin lymphoma[44]. Given that, in many cancer types, chemotherapeutic agents within a single class can be used interchangeably to achieve similar clinical outcomes[45–47], it may be possible to prospectively use these types of data when improving existing regimens or developing new treatment protocols. In patients previously exposed to chemotherapy, knowledge of their altered mutational and clonal landscape could also prompt discussions as to suitability for standard treatment protocols, particularly in the autologous transplant setting, and allow exploring alternative options where appropriate.

## Online content

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

¹Wellcome Sanger Institute, Hinxton, UK. ²Wellcome–MRC Cambridge Stem Cell Institute, Cambridge Biomedical Campus, Cambridge, UK. ³Department of Haematology, University of Cambridge, Cambridge, UK. ⁴Signalling Programme, The Babraham Institute, Babraham Research Campus, Cambridge, UK. ⁵Department of Molecular and Cellular Biology, Baylor College of Medicine, Houston, TX, USA. ⁶Department of Clinical Immunology, Copenhagen University Hospital, Rigshospitalet, Copenhagen, Denmark. ⁷Department of Paediatrics, Cambridge University Hospitals Foundation Trust, Cambridge, UK. ⁸Department of Oncology, Cambridge University Hospitals Foundation Trust, Cambridge, UK. ⁹Oncology R&D, AstraZeneca, Cambridge, UK. ¹⁰Department of Surgery, University of Cambridge, Cambridge, UK. ¹¹Cambridge Biorepository for Translational Medicine, NIHR Cambridge Biomedical Research Centre, University of Cambridge, Cambridge, UK. ¹²Department of Leukemia, Division of Cancer Medicine, MD Anderson Cancer Center, University of Texas, Houston, TX, USA. ¹³York Biomedical Research Institute, Department of Biology, University of York, York, UK. ¹³These authors contributed equally: Emily Mitchell, My H. Pham, Anna Clay, Jyoti Nangalia, Michael R. Stratton. ✉e-mail: jn5@sanger.ac.uk; mrs@sanger.ac.uk

## Methods

### Ethical regulation compliance

Our research complies with all relevant ethical regulations as approved by the National Health Service (NHS) Cambridgeshire 4 Research Ethics Committee, Cambridge East Ethics Committee and the Research Ethics Committee of the University of Texas MD Anderson Cancer Center Institutional Review Board (see additional information below).

### Statistics and reproducibility

No statistical methods were used to predetermine the sample size. We removed a total of 96 colonies from the dataset of 931 previously unpublished colonies: 32 for being technical duplicates, 29 for showing evidence of nonclonality or contamination and 23 due to low coverage. The experiments were not randomized, and the investigators were not blinded to allocation during experiments and outcome assessment.

### Samples

Blood or bone marrow samples from individuals unexposed to chemotherapy were obtained from three sources: (1) STEMCELL Technologies provided frozen mononuclear cells (MNCs) for the cord blood sample that had been collected with written informed consent, including for WGS (catalogue number 70007); all data were previously published. (2) Cambridge Blood and Stem Cell Biobank provided fresh peripheral blood samples taken with written informed consent from two patients at Addenbrooke's Hospital (NHS Cambridgeshire 4 Research Ethics Committee reference 07/MRE05/44 for samples collected before November 2019 and Cambridge East Ethics Committee reference 18/EE/0199 for samples collected from November 2019 onward); all data were previously published. (3) Cambridge Biorepository for Translational Medicine provided frozen bone marrow samples with or without peripheral blood MNCs taken with informed consent from seven deceased organ donors. Samples were obtained at the time of abdominal organ collection (Cambridgeshire 4 Research Ethics Committee reference 15/EE/0152); data were previously published from four individuals, with new data generated from an additional two individuals (PD49236 and PD49327).

Blood samples from individuals previously exposed to chemotherapy were obtained from two sources: (1) Cambridge Blood and Stem Cell Biobank provided fresh peripheral blood samples taken with written informed consent from 22 patients at Addenbrooke's Hospital (NHS Cambridgeshire 4 Research Ethics Committee reference 07/MRE05/44 for samples collected before November 2019 and Cambridge East Ethics Committee reference 18/EE/0199 for samples collected from November 2019 onward); all data were unpublished. One chemotherapy-exposed individual, PD47703, had two samples taken at time points a year apart. All others were sampled at a single time point. (2) Baylor College of Medicine provided single-cell-derived hematopoietic colonies from the bone marrow taken following written informed consent from one patient from MD Anderson Cancer Center (Research Ethics Committee of the University of Texas MD Anderson Cancer Center Institutional Review Board reference PA12-0305 (genomic analysis protocol) and LAB01-473 (laboratory protocol)).

Details of the individuals studied and the samples they provided are listed in Fig. 1a, with additional information in Supplementary Table 1. All participants provided written informed consent to have their anonymized details published. Participants did not receive compensation for taking part in the study.

### Isolation of MNCs from fresh peripheral blood samples

Whole blood was diluted 1:1 with PBS, after which MNCs were isolated using Lymphoprep density gradient centrifugation (STEMCELL Technologies). Red cell lysis was performed on the MNC fraction using one incubation at 4 °C for 15 min with red blood cell lysis buffer (BioLegend).

### Single-cell colony expansion in vitro−liquid culture (unexposed samples)

For all the unexposed normal samples and the PD47703 second-time-point sample, single-cell colony expansion in vitro was undertaken in liquid culture, exactly as previously described[1].

Peripheral blood and cord blood MNC samples underwent CD34+ cell selection using the EasySep human whole blood CD34-positive selection kit (STEMCELL Technologies), with only a single round of magnetic selection. Bone marrow MNCs were not selected for CD34+ cells before sorting.

MNC or CD34-enriched samples were stained (30 min at 4 °C) in PBS/3% FBS containing the following antibodies: CD3 FITC (1 in 500), CD90 PE (1 in 50), CD49f PE-Cy5 (1 in 100), CD38 PE-Cy7 (1 in 100), CD19 A700 (1 in 300), CD34 APC-Cy7 (1 in 100), CD45RA BV421 (1 in 100) and Zombie Aqua (1 in 2,000) (Supplementary Table 8). Cells were then washed and resuspended in PBS/3% FBS for cell sorting. Either a BD Aria III or BD Aria Fusion cell sorter (BD Biosciences) was used to sort 'HSC/MPP' pool cells (Lin−, CD34+, CD38−, CD45RA−) at the National Institute for Health and Care Research (NIHR) Cambridge Biomedical Research Centre (BRC) Cell Phenotyping Hub. Supplementary Fig. 1 illustrates the gating strategy used.

Single phenotypic HSPCs were index-sorted into single wells of 96-well plates containing StemPro medium (STEMCELL Technologies), StemPro Nutrients (0.035%, STEMCELL Technologies), L-glutamine (1%, ThermoFisher), penicillin−streptomycin (1%, ThermoFisher) and cytokines (stem cell factor, 100 ng ml$^{-1}$; FLT3, 20 ng ml$^{-1}$; thrombopoietin, 100 ng ml$^{-1}$; erythropoietin, 3 ng ml$^{-1}$; interleukin-6 (IL-6), 50 ng ml$^{-1}$; IL-3, 10 ng ml$^{-1}$; IL-11, 50 ng ml$^{-1}$; granulocyte−macrophage colony-stimulating factor, 20 ng ml$^{-1}$; IL-2, 10 ng ml$^{-1}$; IL-7, 20 ng ml$^{-1}$; lipids, 50 ng ml$^{-1}$) and expanded into colonies. Cells were incubated at 37 °C, and the colonies that formed were topped up with 50 μl StemPro medium plus supplements at 14 ± 2 days as necessary. At 21 ± 2 days, colonies were collected. DNA extraction was performed using either the DNEasy 96 blood and tissue plate kit (Qiagen) or the Arcturus PicoPure DNA extraction kit (ThermoFisher) per the manufacturer's instructions.

### Single-cell colony expansion in vitro−MethoCult (chemotherapy-exposed samples)

For the chemotherapy-exposed peripheral blood samples, single-cell colonies were expanded in MethoCult (H4435 or H4034, STEMCELL Technologies). MNCs were plated at a density of 7.5−45 × 10$^4$ cells per ml in MethoCult and incubated at 37 °C for 14 days. The cell suspensions were made up in StemSpan II (STEMCELL Technologies) before being mixed thoroughly with MethoCult and plated into a Smart-Dish (STEMCELL Technologies). Individual BFU-E (burst-forming unit erythroid) or CFU-GM (colony-forming unit granulocyte−macrophage) colonies were picked, added to 17 μl proteinase K (PicoPure DNA extraction kit, Fisher Scientific; with each vial of lyophilized proteinase K resuspended in 130 μl reconstitution buffer), and incubated at 65 °C for 6 h and at 75 °C for 30 min to extract DNA in preparation for sequencing.

Previous studies have shown that there is no difference in mutation burden between HSCs and hematopoietic progenitor cells[24] and that there is a mutation burden difference of only approximately 30 SBS mutations between HSCs and mature granulocytes[25].

### WGS of colonies

WGS libraries were prepared from 1−5 ng of extracted DNA from each colony using a low-input enzymatic fragmentation-based library preparation method[48,49]. WGS was performed on the NovaSeq platform (Illumina). Paired-end reads of 150 bp were aligned to the human

reference genome (National Center for Biotechnology Information build 37) using BWA-MEM.

## SBS, indel, structural variant and copy number variant calling

The method for substitution calling involved three main steps: mutation discovery, filtering and genotyping, as described previously[19] (Supplementary Methods).

## Additional variant filtering steps

**Larger dataset containing samples sequenced at a lower sequencing depth.** For the creation of the larger phylogenies (a subset of six chemotherapy-exposed and five unexposed individuals with >40 sequenced colonies), a binomial filtering strategy could be applied as previously described[24] (https://github.com/emily-mitchell/normal_haematopoiesis/2_variant_filtering_tree_building/scripts/).

**Small dataset containing samples sequenced at a relatively high sequencing depth.** For the subset of colonies sequenced at the highest depth (four to ten colonies per individual), we were unable to use binomial filtering approaches due to the low sample number. Instead, the following filters were applied to the data using a custom R script: (1) variants present in more than half the samples from an individual were removed as being most likely germline. (2) Variants were only called as present in a given sample if they had two or more supporting reads and were present at a variant allele fraction (VAF) of ≥0.2 in autosomes or ≥0.4 for sex chromosomes. (3) High- and low-depth sites with a mean depth of >50 or <8 across all samples from an individual were removed.

This variant filtering approach was validated using samples from the normal individuals, in whom both the binomial and nonbinomial filtering strategies were applied to the same samples, giving comparable results (Supplementary Fig. 2). This dataset, comparable across all the individuals in the study, was used for the analysis of SBS and indel mutation burdens.

## Filtering at the colony level

We removed a total of 96 colonies from the dataset of 931 previously unpublished colonies: 32 for being technical duplicates, 29 for showing evidence of nonclonality or contamination and 23 due to low coverage. Visual inspection of the VAF distribution plots was performed, and a peak VAF threshold of <0.4 was used (after the removal of in vitro variants) to identify colonies with evidence of nonclonality.

## Mutation burden analysis

Due to the difficulty in correcting for sequencing depth when only a small number of samples are sequenced per individual, SBS and indel burden analysis was performed on raw data from the subset of chemotherapy-exposed and unexposed samples sequenced at a relatively high depth (four to ten samples per individual; mean coverage 23×, range 13–33×). We have previously shown that sequencing depth has little impact on the SBS mutation burden over this higher range[24]. There were minor differences in sequencing depth when comparing the chemotherapy and normal cohorts or when comparing sequencing depth by chemotherapy exposure, but these would not be expected to affect the interpretation of the results presented (Extended Data Fig. 1).

Given the known influence of age on mutation burden, we built a linear mixed-effect model to quantify how age and chemotherapy treatment influence mutation burdens. Patient ID (PDID) was added as a random effect to the model, which was as follows:

glmer_chemo < − glmer ( round(Number_mutations) ~ Age + Exposure

+ (1|PDID), data = Summary_All, family = poisson(link = "identity")

## Construction of phylogenetic trees

MPBoot, a maximum parsimony tree approximation method[50], was used to build and annotate phylogenetic trees of the relationships between the sampled HSPCs, as previously described[24] (Supplementary Methods).

The key steps to generate the phylogenies shown in Fig. 5 and Extended Data Figs. 8 and 9 are as follows:

1. Generate a 'genotype matrix' of mutation calls for every colony within a donor. Our protocol, based on WGS of single-cell-derived colonies, generates consistent and even coverage across the genome, leading to very few missing values within this matrix (ranging from 0.005 to 0.034 of mutated sites in a given colony across different donors within our cohort). This generates a high degree of accuracy in the constructed trees.

2. Reconstruct phylogenetic trees from the genotype matrix. This is a standard and well-studied problem in phylogenetics. The low fraction of the genome that is mutated in a given colony (<1 per million bases), coupled with the highly complete genotype matrix, means that different phylogenetic methods produce reassuringly concordant trees. We used the MPBoot algorithm for the tree reconstruction, as it proved both accurate and computationally efficient for our dataset.

3. Correct terminal branch lengths for sensitivity to detect mutations in each colony. The trees generated in the previous step have branch lengths proportional to the number of mutations assigned to each branch. For the terminal branches, which contain mutations unique to that colony, variable sequencing depth can underestimate the true numbers of unique mutations, so we correct these branch lengths for the estimated sensitivity to detect mutations based on genome coverage.

4. Make phylogenetic trees ultrametric. After step 3, there is little more than Poisson variation in the corrected mutation burden among colonies from a given donor. As these colonies were all derived from the same time point, we can normalize the branch lengths to have the same overall distance from root to tip (known as an ultrametric tree). We used an 'iteratively reweighted means' algorithm for this purpose.

5. Scale trees to chronological age. As the mutation rate is constant across the human lifespan, we can use it as a 'molecular clock' to linearly scale the ultrametric tree to chronological age.

6. Overlay phenotypic and genotypic information on the tree.

Overlay phenotypic and genotypic information on the tree. The tip of each branch in the resulting phylogenetic tree represents a specific colony in the dataset, meaning that we can depict phenotypic information about each colony underneath its terminal branch (the colored stripes along the bottom of Fig. 5 and Extended Data Figs. 8 and 9). Furthermore, every mutation in the dataset is confidently assigned to a specific branch in the phylogenetic tree. This means that we can highlight branches on which specific genetic events occurred (such as *DNMT3A* or other driver mutations).

To estimate the number of somatic mutations that may have already been acquired by PD37580 by age 13 years (before commencing chlorambucil), we used the linear mixed model defined by Mitchell et al. This model estimates an intercept of 54.57 (that is, the mean number of somatic mutations present at birth), with a slope of 16.832 representing the mean number of somatic mutations acquired each year of life. This results in an expected mean somatic mutation burden of 273 at age 13 years. Assuming that this mutation burden is Poisson distributed provides a 95% prediction interval of 241–306.

## Analysis of driver variants

The variants identified were annotated with VAGrENT (Variation Annotation GENeraTor) (https://github.com/cancerit/VAGrENT) to identify protein-coding mutations and putative driver mutations in each dataset. Supplementary Table 9 lists the 18 genes we have used as our top clonal hematopoiesis genes (those identified in ref. 38 as being

under positive selection in a targeted sequencing dataset of 385 older individuals, with *CHEK2* added as being an additional gene commonly under positive selection in chemotherapy-exposed individuals). 'Oncogenic' mutations (as assessed by E.M.) are shown in Fig. 5 and Extended Data Figs. 8 and 9.

## Bulk cell sorts for NanoSeq sequencing

**Chemotherapy-exposed samples.** MNCs were stained for 30 min at 4 °C in PBS/3% FCS containing the following antibodies: Zombie Aqua (1 in 400), CD3 APC (1 in 80), CD19 AF700 (1 in 80), CD45RA PerCP-Cy5.5 (1 in 80), CCR7 BV711 (1 in 80) and CD14 BV605 (1 in 80). Cells were then washed and resuspended in PBS/3% FBS for sorting. Either a BD Aria III or BD Aria Fusion cell sorter (BD Biosciences) was used to sort various mature cell compartments (B cells, T naive cells, T memory cells and monocytes) at the NIHR Cambridge BRC Cell Phenotyping Hub. For each cell type, approximately 40,000 cells were sorted into Eppendorf tubes containing 50 µl PBS. Further details are provided in Supplementary Table 10 and Supplementary Fig. 3.

**Unexposed normal samples.** MNCs were stained for 30 min at 4 °C in PBS/3% FCS containing the following antibodies: CD3 APC (1 in 80), CD4 BV785 (1 in 80), CD8 BV785 (1 in 40), CD14 BV605 (1 in 80), CD19 AF700 (1 in 80), CD20 PE-Dazzle (1 in 80), CD27 BV421 (1 in 80), CD34 APC-Cy7 (1 in 27), CD38 FITC (1 in 80), CD45RA PerCP-Cy5.5 (1 in 80), CD56 PE (1 in 80), CCR7 BV711 (1 in 80), IgD PE-Cy7 (1 in 100) and Zombie Aqua (1 in 400). Cells were then washed and resuspended in PBS/3% FBS for sorting. Either a BD Aria III or BD Aria Fusion cell sorter (BD Biosciences) was used to sort various mature cell compartments (B cells, CD4+ T naive cells, CD4+ T memory cells, CD8+ T naive cells, CD8+ T memory cells and monocytes) at the NIHR Cambridge BRC Cell Phenotyping Hub. For each cell type, approximately 40,000 cells were sorted into Eppendorf tubes containing 50 µl PBS. Further details are provided in Supplementary Table 11 and Supplementary Fig. 4.

## DNA extraction from bulk mature cell sorts

Approximately 40,000 cells of each mature cell type from the above sorts (suspended in 200 µl PBS) were added to single wells of a 96-well PCR plate and centrifuged. Pellets were resuspended in 17 µl proteinase K (PicoPure DNA extraction kit, Fisher Scientific; with each vial of lyophilized proteinase K resuspended in 130 µl reconstitution buffer) and incubated at 65 °C for 6 h and at 75 °C for 30 min to extract DNA in preparation for sequencing.

## NanoSeq (duplex) sequencing

Extracted DNA (1–5 ng) from bulk cell sorts was submitted to the NanoSeq pipeline for library preparation and sequencing, as has been described previously[25] (Supplementary Methods).

## Mutational signature analysis

The HDP (https://github.com/nicolaroberts/hdp), based on the Bayesian HDP, was used to extract mutational signatures. The HDP was run without priors on SBSs derived from phylogenetic trees obtained from HSPCs and mutations from NanoSeq samples. The NanoSeq mutations were corrected for the trinucleotide context abundance for each sample (Supplementary Methods).

The COSMIC (v3.4)[27] signatures identified were SBS1, SBS5, SBS7a, SBS9 and SBS17. One of the HDP components corresponded to the SBSBlood signature previously reported[30]. Eight components were de novo signatures called predominantly in individuals with chemotherapy exposure. Only signatures with a contribution of >5% of the mutations of the sample's burden were considered.

Eight chemotherapy-related signatures derived from the HDP (SBSA–SBSH) were examined for their occurrence in exposed versus nonexposed individuals. The proportions of the signatures contributing to the samples in the exposed versus nonexposed group were compared using the *t* test for independent samples, with an equal variance assumption. The test was performed in two ways: (1) per cell type (Supplementary Table 5) and (2) by combining the samples across cell types for each individual (Supplementary Table 6). The *P* values were adjusted for multiple testing corrections using the false discovery rate and Bonferroni methods.

Indel signatures were extracted from small indels called from the HSPC dataset using two different methods. First, mSigHdp was run, identifying three distinct indel signatures (https://github.com/steverozen/mSigHdp)[51]. To investigate whether the extracted de novo signatures were composed of reference COSMIC signatures, we used the SigProfilerAssignment decompose tool (https://github.com/AlexandrovLab/SigProfilerAssignment)[52]. Two signatures were successfully decomposed, the first into ID1 and ID2 with a reconstructed cosine similarity of 0.99 (compared to the de novo signature). The second signature was decomposed into ID3, ID5 and ID9 with a reconstructed cosine similarity of 0.93. The final signature, IDA, was initially decomposed into ID2 and ID18 with a reconstructed cosine similarity of 0.88. ID18 is a signature associated with colibactin exposure[19], which is unlikely in this context. This, in combination with the lower cosine similarity and strong support from the mutation spectra of individuals treated with procarbazine, led to this decomposition being rejected.

## Reporting summary

Further information on research design is available in the Nature Portfolio Reporting Summary linked to this article.

## Data availability

Smaller derived datasets needed to perform mutation burden and mutational signature analysis are available on GitHub (https://github.com/emily-mitchell/chemotherapy/) and Zenodo (https://doi.org/10.5281/zenodo.15235476)[53]. Raw sequencing data are available on the European Genome–phenome Archive (accession numbers EGAD00001015339 (WGS dataset) and EGAD00001015340 (NanoSeq dataset)). The main data needed to reanalyze and reproduce the results presented are available on Mendeley Data (https://doi.org/10.17632/2fczcd49yj.1). Source data are provided with this paper.

## Code availability

Code and smaller derived datasets are available on GitHub (https://github.com/emily-mitchell/chemotherapy/) and Zenodo (https://doi.org/10.5281/zenodo.15235476)[53].

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

## Acknowledgements

This work was delivered as part of the Mutographs team supported by the Cancer Grand Challenges partnership funded by Cancer

Research UK (CRUK) (C98/A24032). This work was supported by Wellcome grants 206194 and 220540/Z/20/A and by the NIHR Cambridge Biomedical Research Centre (BRC-1215-555 20014). J.N. was supported by CRUK, Alborda Trust, Rosetrees Trust, Blood Cancer UK and Wellcome. D.J.H. was supported by a fellowship from CRUK (RCCFEL\100072) and received core funding from the CRUK Cambridge Centre (A25117). E.L. was supported by a Wellcome–Royal Society Sir Henry Dale Fellowship (107630/Z/15/Z). E.L. and D.J.H. were supported by core support grants from Wellcome and the Medical Research Council (MRC) to the Wellcome–MRC Cambridge Stem Cell Institute (203151/Z/16/Z). Samples were provided by the Cambridge Blood and Stem Cell Biobank, which is supported by the NIHR Cambridge BRC, Wellcome Trust–MRC Stem Cell Institute and the Cambridge Experimental Cancer Medicine Centre, UK. This research was supported by the NIHR Cambridge BRC Cell Phenotyping Hub. We are grateful to the donors, donor families and the Cambridge Biorepository for Translational Medicine for the gift of their tissue. We would like to thank L. O'Neill, K. Roberts, K. Smith, S. Austin-Guest and the staff of DNA Pipelines at the Wellcome Sanger Institute for their contributions. For the purpose of open access, we have applied a CC BY public copyright license to any author-accepted paper version arising from this submission. The funders had no role in study design, data collection and analysis, decision to publish or preparation of the manuscript.

## Author contributions

A. Clay and E.M. performed cell sorting and culture. E.M. performed mutation calling and phylogeny construction. M.H.P. performed mutational signature analysis and prepared figures. R.S. and R.R. supported mutational signature analysis. N.W., M.S.C. and D.L. supported phylogenetic and signature analysis. S.P., H.E.M., D.K. and E.L. supported cell isolation and culture. H.J., S.M., E.D. and J.W. supported mutation analysis. A. Cagan helped with figure design. U.M., G.J.D., K.M., A.S., A.V., G.F., D.J.H., K.S.P., K.T. and E.J.B. provided clinical advice and patient samples. P.J.C. and I.M. provided scientific advice. J.I.H. and M.A.G. provided clinical samples and supported cell culture and mutation identification. L.H. provided project coordination. N.F.Ø. provided development of laboratory methods. E.M., J.N. and M.R.S. wrote the paper. J.N. and M.R.S. directed the research.

## Competing interests

G.J.D. and U.M. are employees of and shareholders in AstraZeneca. M.R.S. and P.J.C. are cofounders of and shareholders in Quotient Therapeutics. The other authors declare no competing interests.

## Additional information

**Extended data** is available for this paper at https://doi.org/10.1038/s41588-025-02234-x.

**Correspondence and requests for materials** should be addressed to Jyoti Nangalia or Michael R. Stratton.

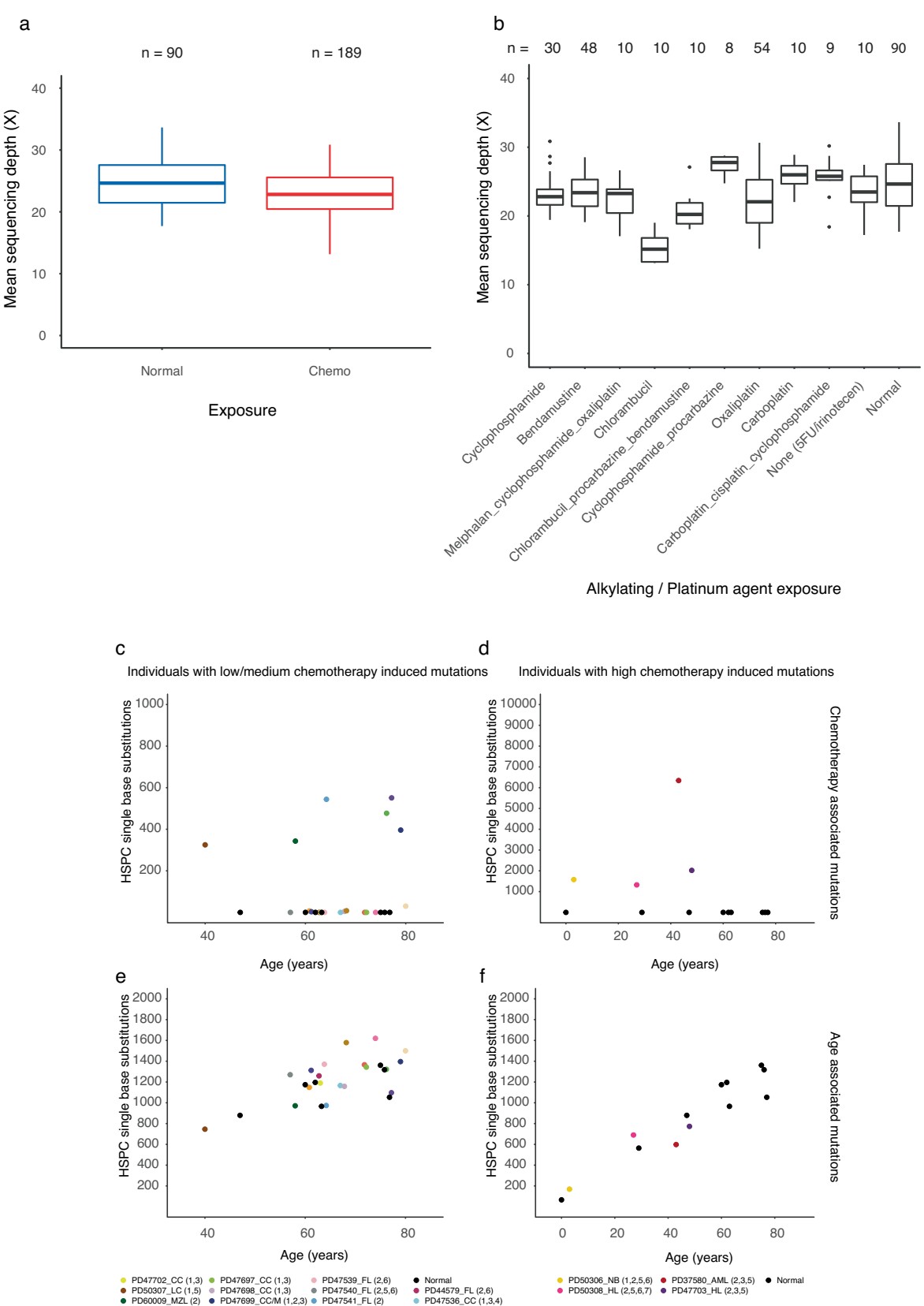

**Extended Data Fig. 1 | See next page for caption.**

**Extended Data Fig. 1 | Mean sequencing depth in the normal and chemotherapy exposed HSPC colonies used for mutation burden analysis, and patterns of mutation accumulation. a**, Box plot representing the quartile distribution of mean sequencing depth in 90 colonies from normal (blue) and 189 colonies from chemotherapy exposed (red) individuals. The boxes indicate the median and interquartile range, the whiskers denote the minimum and maximum, with outlying values represented as points. **b**, Boxplot comparing the mean sequencing depth between Alkylating/ Platinum agent exposed and non-exposed colonies. The number of colonies in each agent group are shown at the top of the plot. The boxes indicate the median and interquartile range, the whiskers denote the minimum and maximum, with outlying values represented as points. **c-f**, HSC single base substitutions associated with chemotherapy (**c,d**) and age (**e,f**). CC, colorectal carcinoma; LC, lung cancer; NB, neuroblastoma; FL, follicular lymphoma; DLBL, diffuse large B cell lymphoma; MZL, marginal zone lymphoma; LL, lymphoplasmacytic lymphoma; M, multiple myeloma; HL, Hodgkin lymphoma; AML, acute myeloid leukaemia; 1, Platinum agents; 2,Alkylating agents; 3,Antimetabolites; 4,Topo I inhibitors; 5,Topo II inhibitors; 6,Vinca alkaloids; 7,Cytotoxic antibiotics.

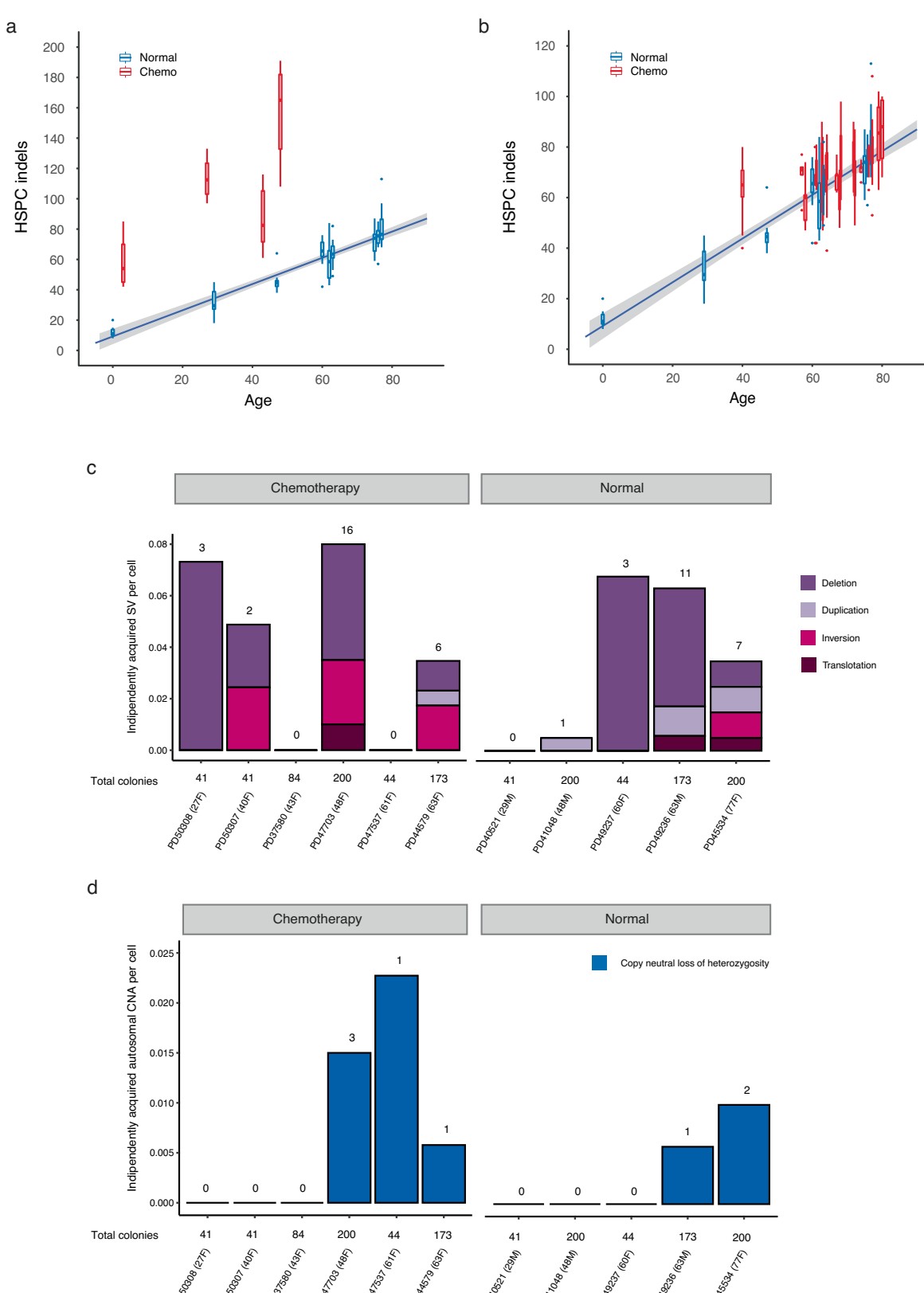

**Extended Data Fig. 2 | See next page for caption.**

**Extended Data Fig. 2 | Indel mutational burden in normal and chemotherapy exposed HSPCs.** Burden of small indels in single HSPC colonies with age (years) across normal (blue) and the four chemotherapy exposed (red) individuals with the highest indel burdens. The points represent individual HSPC colonies. The boxes indicate the median and interquartile range, the whiskers denote the minimum and maximum. The blue line represents a regression of age on mutation burden, with 95% CI shaded. **b**, Depiction of data as in a, but the y-axis is cut off at 120 indels for better visualisation of the majority of the chemotherapy-exposed data. The points represent individual HSPC colonies. The boxes indicate the median and interquartile range, the whiskers denote the minimum and maximum. The blue line represents a regression of age on mutation burden, with 95% CI shaded **c,d**, Bar plots showing the number of structure variant types (**c**) and the number of the number of independently acquired autosomal copy number aberrations (CNAs) (**d**) in each individual from chemotherapy and normal groups. The absolute number of events found in each individual is shown at the top of each bar. Individuals and the total number of isolated colonies are sorted by age within each group.

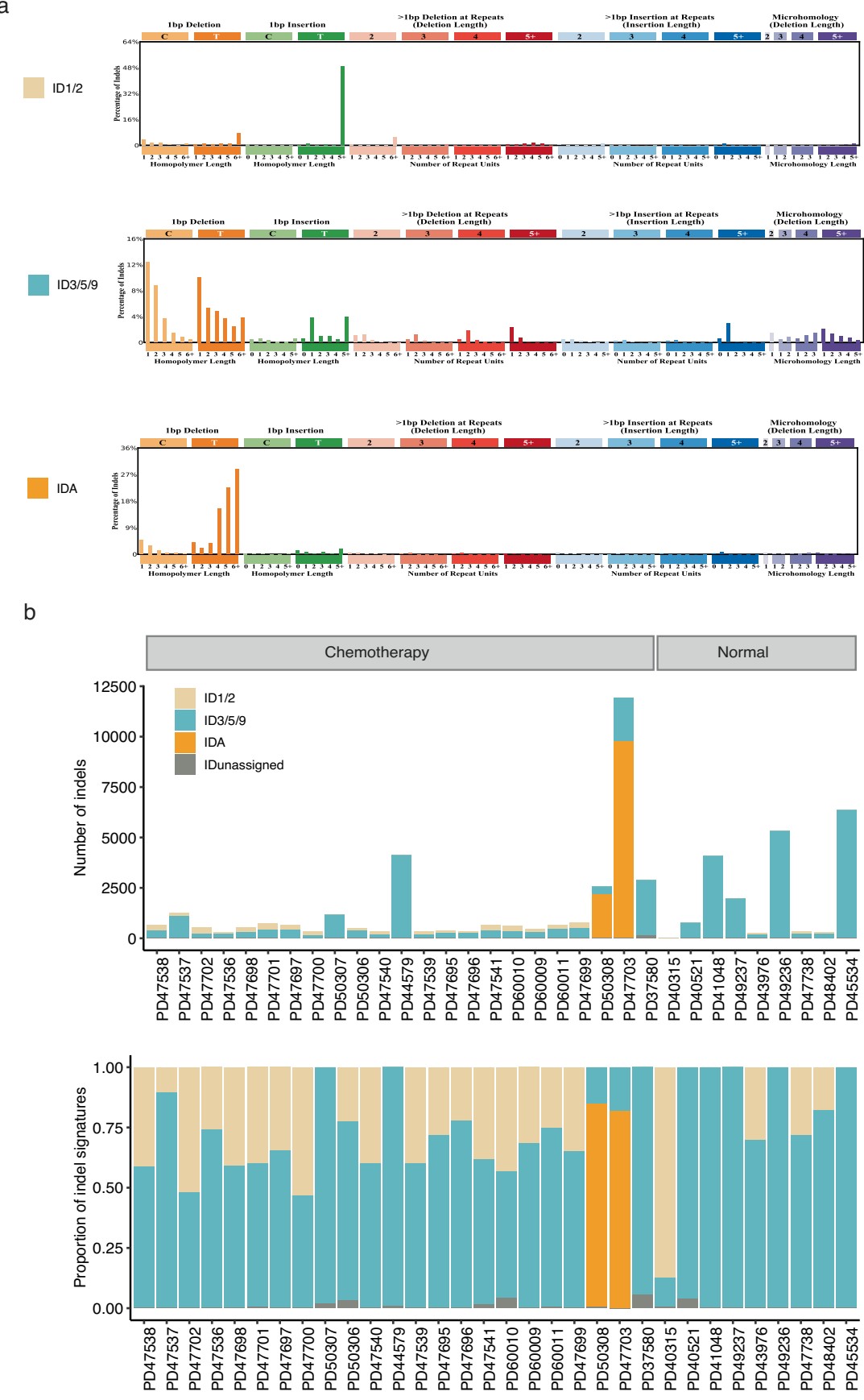

**Extended Data Fig. 3 | See next page for caption.**

**Extended Data Fig. 3 | Indel signatures that are present in normal and chemotherapy exposed blood. a**, Three indel signatures (ID1/2, ID3/5/9, IDA) were extracted by sigHDP. The context on the x-axis show the contributions of different types of indels, grouped by whether variants are deletions or insertions, the size of the event, the presence within repeat units and the sequence content of the indel. **b**, The proportion of indels and indels burden per mutational signatures across 22 chemotherapy exposed and 9 normal individuals, extracting using msigHDP (Methods). Each column represents samples from one individual. Signatures with the contribution <5% are considered as 'unassigned'.

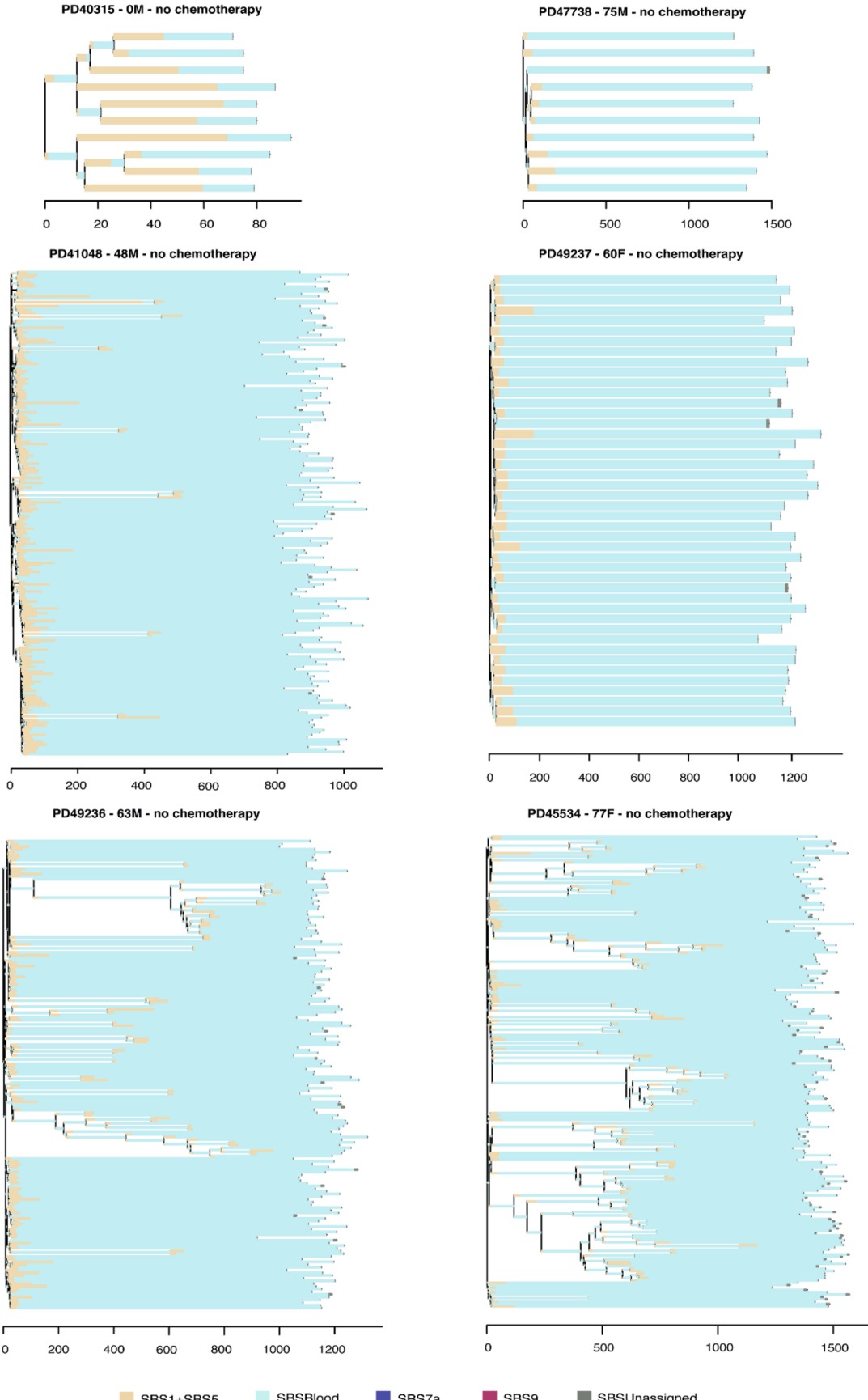

**Extended Data Fig. 4 | Phylogenetic trees and mutational signatures in normal individuals.** Branch lengths correspond to SBS burdens. A stacked bar plot represents the SBS mutational signatures contributing to each branch with color code below the trees. SBSUnassigned indicates mutations that could not confidently be assigned to any reported signature.

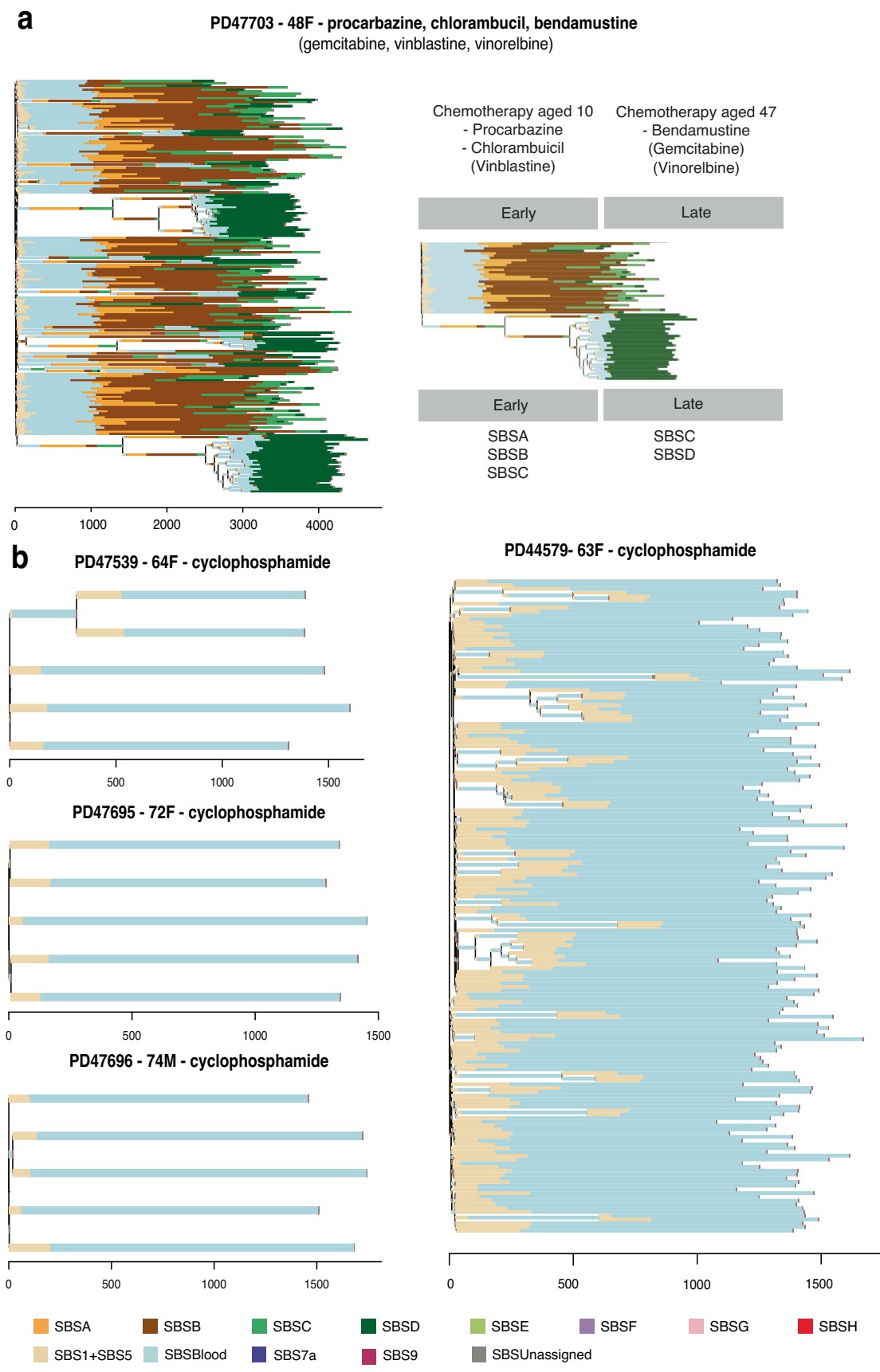

**a** PD47703 - 48F - procarbazine, chlorambucil, bendamustine
(gemcitabine, vinblastine, vinorelbine)

Chemotherapy aged 10
- Procarbazine
- Chlorambuicil
(Vinblastine)

Chemotherapy aged 47
- Bendamustine
(Gemcitabine)
(Vinorelbine)

Early | Late

Early | Late

SBSA | SBSC
SBSB | SBSD
SBSC

**b** PD47539 - 64F - cyclophosphamide

PD44579- 63F - cyclophosphamide

PD47695 - 72F - cyclophosphamide

PD47696 - 74M - cyclophosphamide

SBSA  SBSB  SBSC  SBSD  SBSE  SBSF  SBSG  SBSH
SBS1+SBS5  SBSBlood  SBS7a  SBS9  SBSUnassigned

**Extended Data Fig. 5 | See next page for caption.**

**Extended Data Fig. 5 | Phylogenetic trees and mutational signatures in individuals treated with alkylating agents. a**, Phylogenetic tree of 48-year-old chemotherapy exposed female (PD47703). Branch lengths correspond to SBS burdens. A stacked bar plot represents the SBS mutational signatures contributing to each branch with colour code below the trees. SBSUnassigned indicates mutations that could not confidently be assigned to any reported signature. She had been treated with chlorambucil and procarbazine at age 10 (early), and bendamustine at age 47 (late). **b**, Phylogenetic trees and SBS mutational signatures in individuals treated with cyclophosphamide.

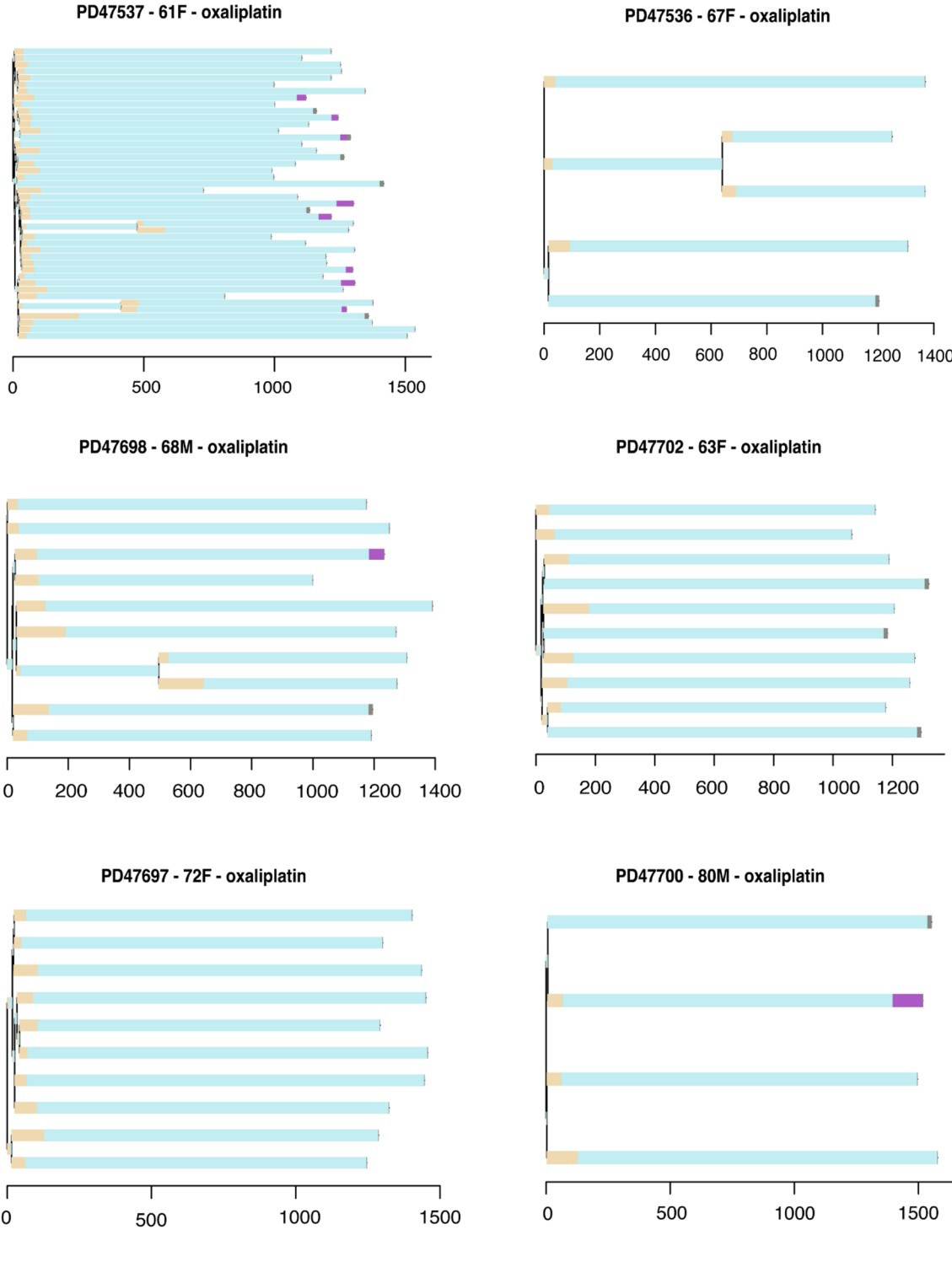

**Extended Data Fig. 6 | Phylogenetic trees and mutational signatures in individuals treated with oxaliplatin.** Branch lengths correspond to SBS burdens. A stacked bar plot represents the SBS mutational signatures contributing to each branch with colour code below the trees. SBSUnassigned indicates mutations that could not confidently be assigned to any reported signature.

a

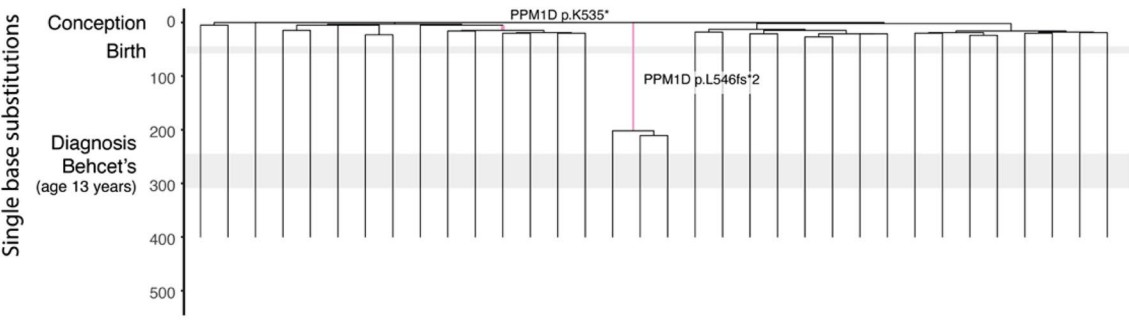

**PD37580 43 year female - phylogeny truncated at 400 SBSs to allow better visualistion of early life**
**Long-term chlorambucil** given to treat Behcet's disease diagnosed aged 13
Cytarabine, daunorubicin, mitxantrone, etoposide aged 43

Timing two early *PPM1D* drivers to before Behcet's diagnosis

b

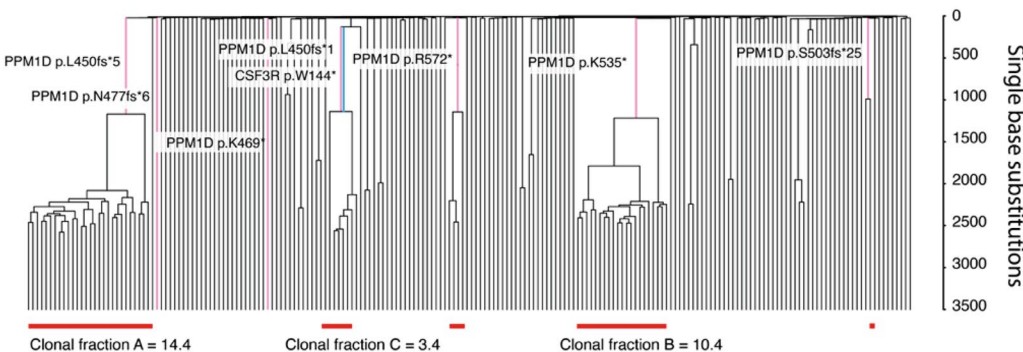

**PD47703 - 48 year chemotherapy exposed female - 200 peripheral blood HSPC colonies**
Chlorambucil, vinorelbine, procarbazine aged 10
Bendamustine, gemcitabine, vinorelbine aged 47

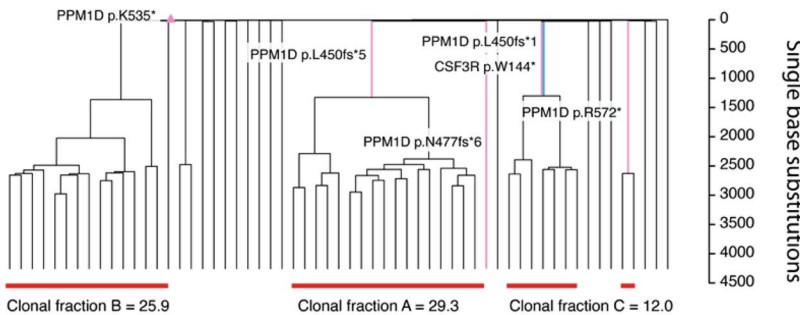

**PD47703 - 49 year chemotherapy exposed female (repeat sampling 1 year later) - 59 peripheral blood HSPC colonies**
Chlorambucil, vinorelbine, procarbazine aged 10
Bendamustine, gemcitabine, vinorelbine aged 47
Cyclophosphamide, Doxorubicin, Vincristine aged 48

**Extended Data Fig. 7 | Annotated HSPC phylogenies for two chemotherapy treated individuals.** Phylogenies were constructed for PD37580 (**a**) and PD47703 (**b**) individuals using shared mutation data and the algorithm MPBoot (Methods). In all phylogenies, branch lengths reflect the number of SBS mutations assigned to the branch. The y-axis represents the number of SBSs accumulating over time. Each tip on a phylogeny represents a single colony. Chemotherapy agents and the age of exposure to them are shown on top of the trees. **a**, PD37580 phylogeny of early life, truncated at 400 SBS mutations to allow better visualisation of the timing of acquisition of two early *PPM1D* mutations (pink). The number of mutations at age 13 was estimated using the linear mixed model described in Mitchell et al with 95% CI based on mutation burden being Poisson distributed

as described in methods (241-306 single base subsitutions). **b**, Comparison of phylogenies created from peripheral blood samples taken from PD44703 one year apart. Pathogenic mutations in *PPM1D* have been highlighted (pink) to facilitate comparison of clone sizes at each timepoint. In addition a loss of function mutation in *CSF3R* has been highlighted (blue), which could also be contributing to loss of haematopoietic reserve and cytopenias. Red bars show the size of clonal fractions at each timepoint. Terminal branches have been adjusted for sequence coverage, and overall root-to-tip branch lengths have been normalized to the same total length (because all colonies were collected from a single time point).

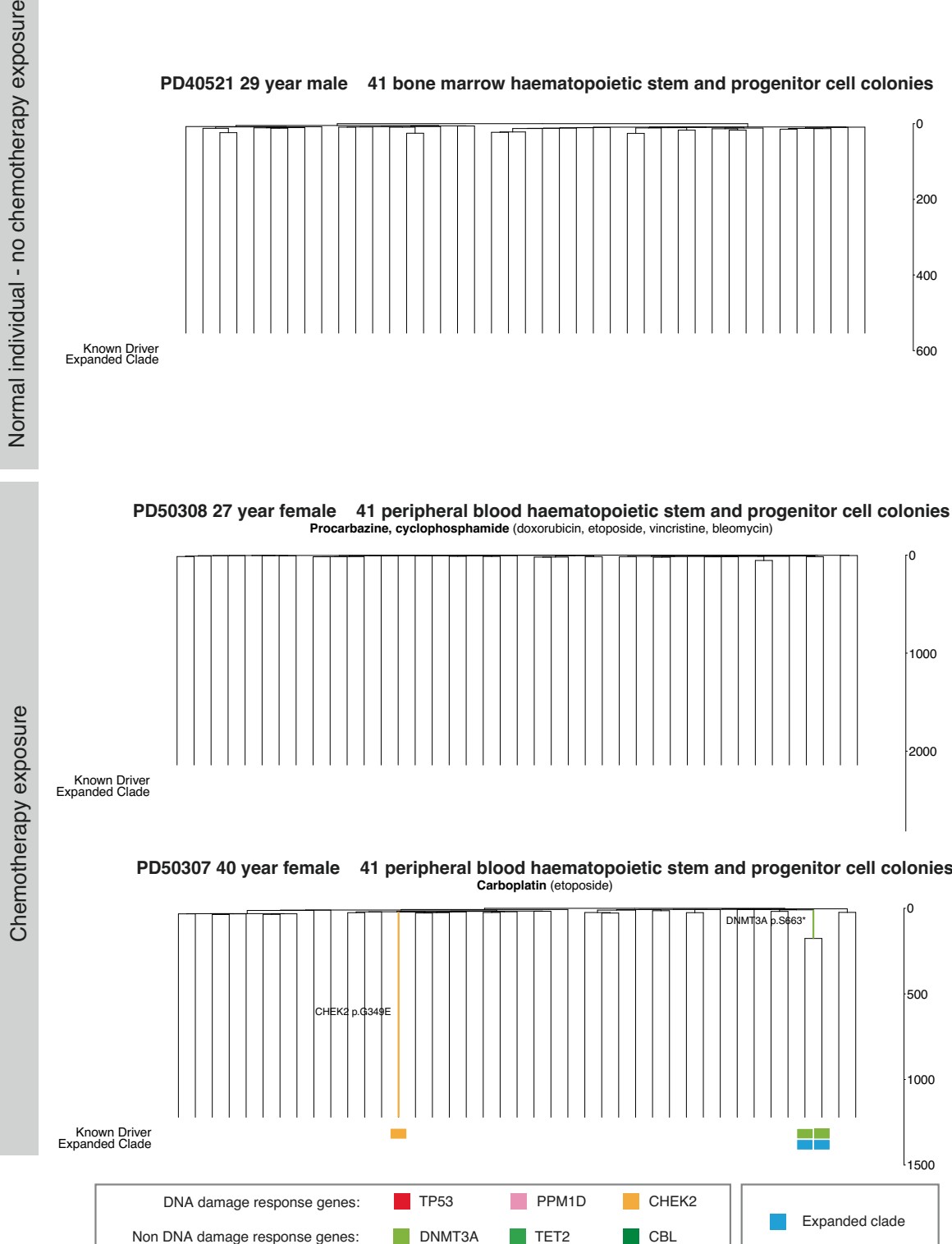

**Extended Data Fig. 8 | See next page for caption.**

**Extended Data Fig. 8 | HSPC phylogenies for three young adult individuals.**
Phylogenies for one normal young adult individual (top) and two young adult chemotherapy-treated individuals (bottom) were constructed using shared mutation data and the algorithm MPBoot (Methods). Branch lengths reflect the number of mutations assigned to the branch with terminal branches adjusted for sequence coverage, and overall root-to-tip branch lengths have been normalised to the same total length (because all colonies were collected from a single time point). The y-axis represents the number of SBSs accumulating over time. Each tip on a phylogeny represents a single colony, with the respective numbers of colonies of each cell and tissue type recorded at the top. Onto these trees, we have layered clone and colony-specific phenotypic information. We have highlighted branches on which we have identified known oncogenic drivers in one of 18 clonal haematopoiesis genes (Supplementary Table 2) colour-coded by gene. A heat map at the bottom of each phylogeny highlights colonies from known driver clades coloured by gene, and the expanded clades (defined as those with a clonal fraction above 1%) in blue. Regarding the treatment of PD50307 donor, carboplatin was administered via intravenous infusion on Day 1, followed by Etoposide on the same day. Subsequently, the patient received oral doses of Etoposide on Days 2 and 3.

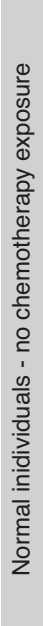

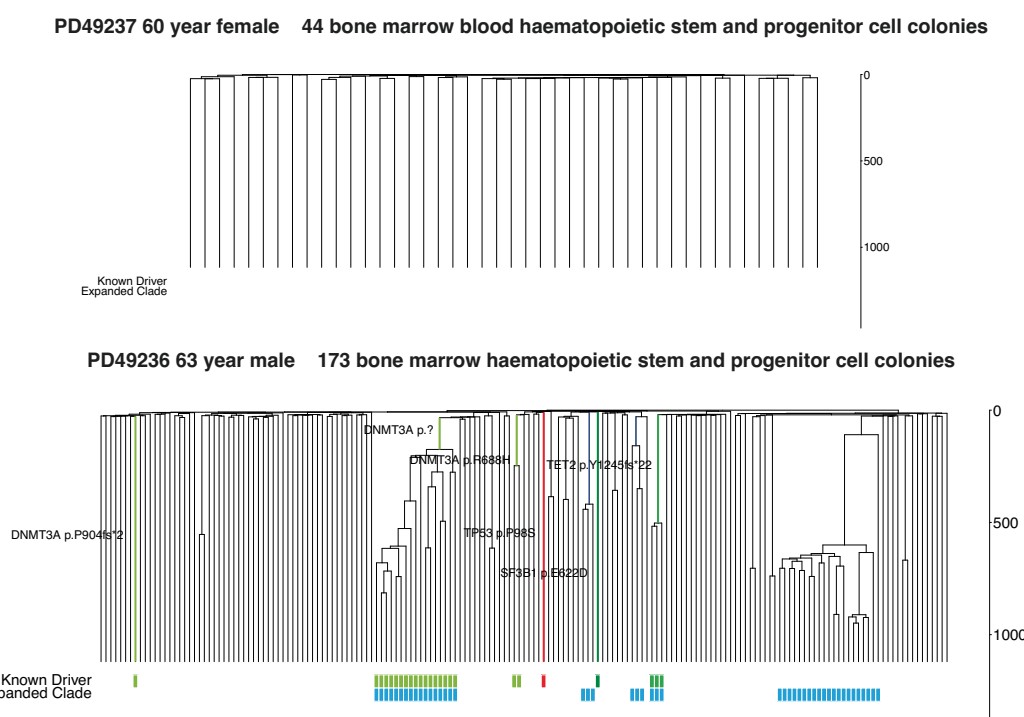

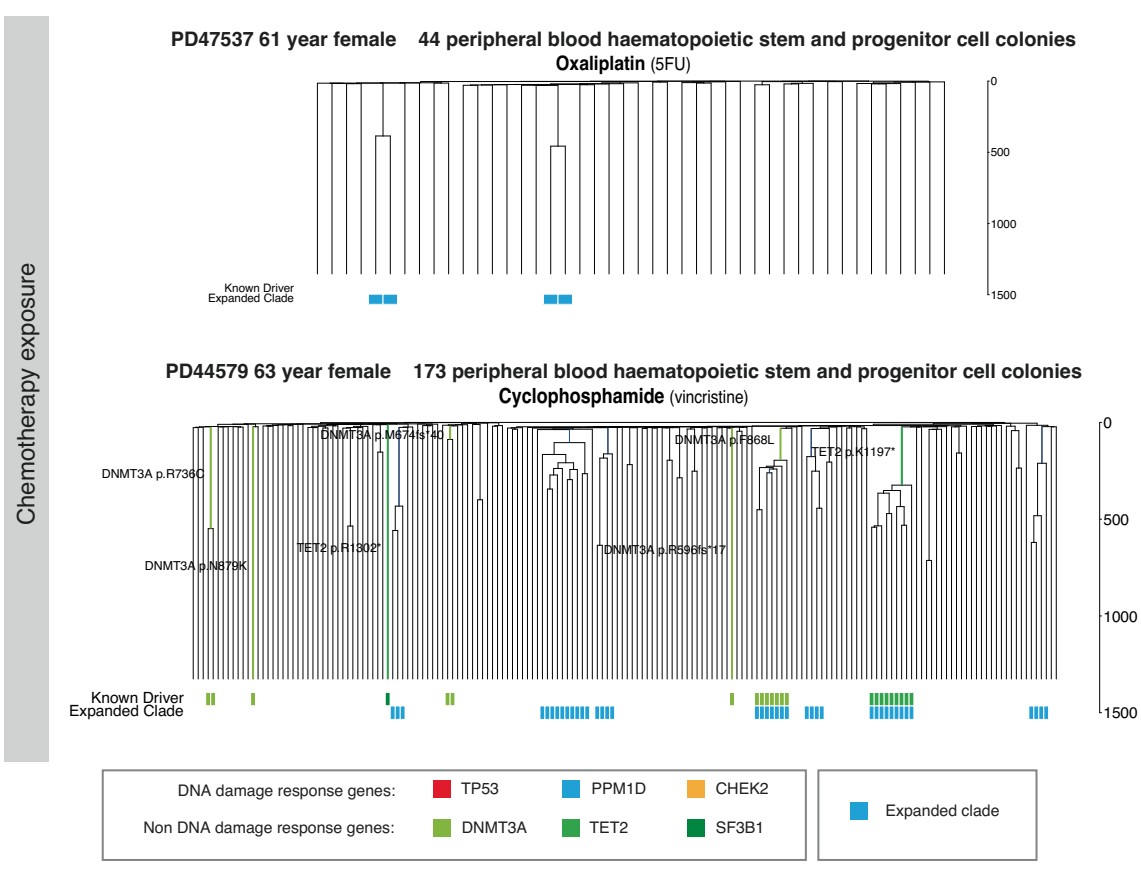

**Extended Data Fig. 9 | See next page for caption.**

**Extended Data Fig. 9 | HSPC phylogenies for four older adult individuals.**
Phylogenies for two normal individuals (top) and two chemotherapy-treated individuals (bottom) were constructed using shared mutation data and the algorithm MPBoot (Methods). Branch lengths reflect the number of mutations assigned to the branch with terminal branches adjusted for sequence coverage, and overall root-to-tip branch lengths have been normalised to the same total length (because all colonies were collected from a single time point). The y-axis represents the number of SBSs accumulating over time. Each tip on a phylogeny represents a single colony, with the respective numbers of colonies of each cell and tissue type recorded at the top. Onto these trees, we have layered clone and colony-specific phenotypic information. We have highlighted branches on which we have identified known oncogenic drivers in one of 18 clonal haematopoiesis genes (Supplementary Table 2) colour-coded by gene. A heat map at the bottom of each phylogeny highlights colonies from known driver clades coloured by gene, and the expanded clades (defined as those with a clonal fraction above 1%) in blue.

# Reporting Summary

## Statistics

For all statistical analyses, confirm that the following items are present in the figure legend, table legend, main text, or Methods section.

| n/a | Confirmed | |
|---|---|---|
| ☐ | ☒ | The exact sample size (*n*) for each experimental group/condition, given as a discrete number and unit of measurement |
| ☐ | ☒ | A statement on whether measurements were taken from distinct samples or whether the same sample was measured repeatedly |
| ☐ | ☒ | The statistical test(s) used AND whether they are one- or two-sided *Only common tests should be described solely by name; describe more complex techniques in the Methods section.* |
| ☐ | ☒ | A description of all covariates tested |
| ☐ | ☒ | A description of any assumptions or corrections, such as tests of normality and adjustment for multiple comparisons |
| ☐ | ☒ | A full description of the statistical parameters including central tendency (e.g. means) or other basic estimates (e.g. regression coefficient) AND variation (e.g. standard deviation) or associated estimates of uncertainty (e.g. confidence intervals) |
| ☐ | ☒ | For null hypothesis testing, the test statistic (e.g. *F*, *t*, *r*) with confidence intervals, effect sizes, degrees of freedom and *P* value noted *Give P values as exact values whenever suitable.* |
| ☒ | ☐ | For Bayesian analysis, information on the choice of priors and Markov chain Monte Carlo settings |
| ☒ | ☐ | For hierarchical and complex designs, identification of the appropriate level for tests and full reporting of outcomes |
| ☒ | ☐ | Estimates of effect sizes (e.g. Cohen's *d*, Pearson's *r*), indicating how they were calculated |

*Our web collection on statistics for biologists contains articles on many of the points above.*

## Software and code

Policy information about availability of computer code

| Data collection | None. |
|---|---|
| Data analysis | Open source programs used (also stated in manuscript): List of programs and softwares: • R: version 3.6.1 • BWA-MEM: version 0.7.17/0.7.5a-r405 (https://sourceforge.net/projects/bio-bwa/) • cgpCaVEMan: version 1.11.2/1.13.14/1.14.1 (https://github.com/cancerit/CaVEMan) • cgpPindel: version 2.2.5/3.2.0/3.3.0 (https://github.com/cancerit/cgpPindel) • Brass: version 6.1.2/6.2.0/6.3.0/6.3.4 (https://github.com/cancerit/BRASS) • ASCAT NGS: version 4.2.1/4.3.3 (https://github.com/cancerit/ascatNgs) • VAGrENT: version 3.5.2/3.6.0/3.6.1 (https://github.com/cancerit/VAGrENT) • GRIDSS: version 2.9.4 (https://github.com/PapenfussLab/gridss) • MPBoot: version 1.1.0 (https://github.com/diepthihoang/mpboot) • cgpVAF: version 2.4.0 (https://github.com/cancerit/vafCorrect) • FlowJo: version 10 • HDP: (https://github.com/nicolaroberts/hdp) • bcftools: version 1.18 (https://github.com/samtools/bcftools) • VerifyBamID2: version 2.0.1 (https://anaconda.org/bioconda/verifybamid2) • mSigHDP: version 2.1.2 (https://github.com/steverozen/mSigHdp) • SigProfilerAssignment: version 0.1.0 (https://github.com/AlexandrovLab/SigProfilerAssignment) |

For manuscripts utilizing custom algorithms or software that are central to the research but not yet described in published literature, software must be made available to editors and reviewers. We strongly encourage code deposition in a community repository (e.g. GitHub). See the Nature Portfolio guidelines for submitting code & software for further information.

## Data

Policy information about availability of data

All manuscripts must include a data availability statement. This statement should provide the following information, where applicable:

- Accession codes, unique identifiers, or web links for publicly available datasets
- A description of any restrictions on data availability
- For clinical datasets or third party data, please ensure that the statement adheres to our policy

Sequence data that support the findings of this study have been deposited in the European Genome-Phenome Archive (https://www.ebi.ac.uk/ega/home). Additional data is available on github (https://github.com/emily-mitchell/chemotherapy/).  Raw sequencing data is available on EGA (accession number WGS dataset EGAD00001015339 and Nanoseq dataset EGAD00001015340). The main data needed to reanalyse / reproduce the results presented is available on Mendeley Data (DOI: 10.17632/2fczcd49yj.1).

Publically available datasets used:
Human reference genome (NCBI build37)

## Research involving human participants, their data, or biological material

Policy information about studies with human participants or human data. See also policy information about sex, gender (identity/presentation), and sexual orientation and race, ethnicity and racism.

| | |
|---|---|
| Reporting on sex and gender | Information on age and sex  is included for all individuals  in Supplementary table 1 and Fig. 1a. |
| Reporting on race, ethnicity, or other socially relevant groupings | Information on race and ethnicity have not been reported / provided  in this study. |
| Population characteristics | All relevant information about donors is provided in Supplementary table 1, which includes information on age, sex, diagnoses, treatment regimens, time since exposure to chemotherapy, any exposure to radiotherapy.  The number of cycles of each chemotherapy is also provided. |
| Recruitment | 22 chemotherapy exposed participants were recruited from oncology / haematology clinics at Addenbrooke's Hospital Cambridge, with the only inclusion criteria being that they had been exposed to chemotherapy.  One chemotherapy exposed participant was recruited from MD Anderson Cancer Centre. 7 unexposed normal individuals had been recruited to a previously published study (Mitchell et al 2022). Two additional normal donors were recruited from Cambridge Biorepository for Translational Medicine with no specific inclusion criteria other than that they had not been previously exposed to chemotherapy. |
| Ethics oversight | Blood or bone marrow samples from individuals un-exposed to chemotherapy were obtained from three sources: 1) Stem Cell Technologies provided frozen mononuclear cells (MNCs) for the cord blood sample that had been collected with informed consent, including for whole genome sequencing (catalog #70007); all data previously published. 2) Cambridge Blood and Stem Cell Biobank (CBSB) provided fresh peripheral blood samples taken with informed consent from two patients at Addenbrooke's Hospital (NHS Cambridgeshire 4 Research Ethics Committee reference 07/MRE05/44 for samples collected pre-November 2019 and Cambridge East Ethics Committee reference 18/EE/0199 for samples collected from November 2019 onwards; all data previously published. 3) Cambridge Biorepository for Translational Medicine (CBTM) provided frozen bone marrow +/- peripheral blood MNCs taken with informed consent from seven deceased organ donors. Samples were collected at the time of abdominal organ harvest (Cambridgeshire 4 Research Ethics Committee reference 15/EE/0152); data previously published from 4 individuals with new data generated from an additional 2 individuals (PD49236 and PD49327).<br><br>Blood samples from individuals previously exposed to chemotherapy were obtained from two sources: 1) Cambridge Blood and Stem Cell Biobank (CBSB) provided fresh peripheral blood samples taken with informed consent from 22 patients at Addenbrooke's Hospital (NHS Cambridgeshire 4 Research Ethics Committee reference 07/MRE05/44 for samples collected pre-November 2019 and Cambridge East Ethics Committee reference 18/EE/0199 for samples collected from November 2019 onwards; all unpublished data. One chemotherapy exposed individual, PD44703, had two samples taken at timepoints a year apart. All others were sampled at a single timepoint. 2) Baylor College of Medicine provided single cell-derived haematopoietic colonies from bone marrow taken following informed consent from 1 patient from MD Anderson Cancer Centre; Research Ethics Committee of the University of Texas MD Anderson Cancer Centre Institutional Review Board reference PA12-0305 (genomic analysis protocol) and LAB01-473 (laboratory protocol). |

Note that full information on the approval of the study protocol must also be provided in the manuscript.

# Field-specific reporting

Please select the one below that is the best fit for your research. If you are not sure, read the appropriate sections before making your selection.

☒ Life sciences   ☐ Behavioural & social sciences   ☐ Ecological, evolutionary & environmental sciences

For a reference copy of the document with all sections, see nature.com/documents/nr-reporting-summary-flat.pdf

# Life sciences study design

All studies must disclose on these points even when the disclosure is negative.

| | |
|---|---|
| Sample size | We optimised the number of chemotherapy exposed individuals (23) and number of haematopoietic stem cells sequenced at higher depth per individual (4-10) to describe the mutation burden and mutational signatures in haematopoietic stem and progenitor cells across a range of chemotherapy exposures.  Duplex sequencing was used to allow interrogation of the same information in mature blood cell subsets for 18 chemotherapy exposed individuals. In addition for a subset of six chemotherapy exposed individuals and five normal individuals, we sequenced larger numbers of HSPC colonies to provide a larger dataset for mutational signature analysis and describe changes in clonal structure with chemotherapy exposure.  No power calculation was performed, and there was no target effect size. No sample size calculation was performed. |
| Data exclusions | Per pre-established criteria, genomes with a sequencing depth of less than 7X (23 samples) or with a VAF distribution showing evidence of non-clonality or contamination (peak VAF < 40%) (29 samples) were excluded from the analysis. |
| Replication | While the specific donor samples used have been exhausted, the results from this study should be generally reproducible in separate individuals of the same age and chemotherapy exposures, using the protocols and code included in this manuscript. For a single individual the haematopoietic stem and progenitor cell phylogeny was reconstructed from samples taken at two timepoints one year apart and show reproducible results. In some ways each individual exposed to an chemotherapeutic agent represents a an experimental replication: oxaliplatin n=8; carboplatin n = 2; cyclophosphamide n = 8; bendamustine n = 5; chlorambucil n = 2; 5FU/capecitabine n = 9; irinotecan n=5; doxorubicin n = 4; etoposide n = 4; vincristine n = 7. All other agents included were only received by a single individual. |
| Randomization | This is not relevant to our study. All individuals were haematopoietically normal, and there was no test versus control groups. |
| Blinding | Blinding was not relevant to our study. There was no test performed that required blinding. |

# Reporting for specific materials, systems and methods

We require information from authors about some types of materials, experimental systems and methods used in many studies. Here, indicate whether each material, system or method listed is relevant to your study. If you are not sure if a list item applies to your research, read the appropriate section before selecting a response.

## Materials & experimental systems

| n/a | Involved in the study |
|---|---|
| ☐ | ☒ Antibodies |
| ☒ | ☐ Eukaryotic cell lines |
| ☒ | ☐ Palaeontology and archaeology |
| ☒ | ☐ Animals and other organisms |
| ☒ | ☐ Clinical data |
| ☒ | ☐ Dual use research of concern |
| ☒ | ☐ Plants |

## Methods

| n/a | Involved in the study |
|---|---|
| ☒ | ☐ ChIP-seq |
| ☒ | ☐ Flow cytometry |
| ☒ | ☐ MRI-based neuroimaging |

## Antibodies

| | |
|---|---|
| Antibodies used | Marker; Fluorochrome; Manufacturer;  Catalogue Number; Clone; Dilution; Citation<br>CD3; FITC; BD;  555339; HIT3a; 1 in 500; Beverley PC et al. Eur J Immunol. 1981; 11(4):329-334.<br>CD90; PE; Biolgend;  328110; 5E10; 1 in 50; Adutler-Lieber S, et al. 2013. J Cardiovasc Pharmacol Therap. 18:78.<br>CD49f; PECy5; BD;  551129; GoH3; 1 in 100; Aumailley et al. Exp Cell Res. 1990; 188(1):55-60.<br>CD19; A700; Biolgend;  302226; HIB19; 1 in 300; Boyle M, et al. 2015. J Infect Dis. 212: 416-425.<br>CD34; APCCy7; Biolgend;  343514; 581; 1 in 100; Bigley V, et al. 2011. J Exp Med. 208:227.<br>Zombie ; Aqua; Biolgend;  423101; NA; 1 in 2000; Berg J, et al. 2013. J Exp Med. 210:2803.<br>CD38; PECy7; Biolgend;  303516; HIT2; 1 in 100; Chaimowitz N, et al. 2011. J Immunol. 187:5114.<br>CD45RA; BV421; Biolgend;  304130; HI100; 1 in 100; Causi E, et al. 2015. PLoS One. 10: 0136717.<br><br>Marker; Fluorochrome; Manufacturer;  Catalogue Number; Clone; Dilution; Citation<br>Zombie; Aqua; biolegend;  423101; NA; 1 in 400; Berg J, et al. 2013. J Exp Med. 210:2803. |

CD3; APC; biolegend;  300301; HIT3A; 1 in 80; Kaushal A, et al. 2021. Blood Cancer Discov. 2:600
CD4; BV785; biolegend;  317401; OKT4; 1 in 80; Jung IY, et al. 2022. Sci Transl Med. 14:eabn7336.
CD8 ; BV785; biolegend;  301045; RPA-T8; 1 in 40; Swadling L, et al. 2020. Cell Rep. 30:687.
CD14; BV605; biolegend;  367125; 63D3; 1 in 80; Perry JSA, et al. 2018. Immunity. 48:923.
CD19; AF700; biolegend;  302225; HIB19; 1 in 80; Viny AD, et al. 2019. Cell Stem Cell. 25:682
CD20; PE Dazzle; biolegend;  302347; 2H7; 1 in 80; Brouwer PJM, et al. 2020. Science. 369:643
CD27; BV421; biolegend;  356429; M-T271; 1 in 80; Tran TM, et al. 2020. Immunity. 51(4):750-765.
CD34; Apc-Cy7; biolegend;  343513; 581; 1 in 27; Takayama N, et al. 2020. Cell Stem Cell. 28(3):488-501.e10
CD38; FITC; biolegend;  980304; HIT2; 1 in 80; NA
CD45RA; PerCPCy5.5; biolegend;  304121; HI100; 1 in 80; Sammicheli S, et al. 2012. J Autoimmun. 38:304
CD56; PE; biolegend;  355503; 39D5; 1 in 80; de Andrade LF, et al. 2019. JCI Insight. 4:e133103
CCR7; BV711; biolegend;  353227; G043H7; 1 in 80; Arunachalam PS, et al. 2021. Nature. 596:410.
IgD; PECy7; biolegend;  348209; IA6-2; 1 in 100; Rouers A, et al. 2021. Cell Rep Med. 2:100278

Marker; Fluorochrome; Manufacter;  Catalogue Number; Clone; Dilution; Citation
Zombie; Aqua; biolegend;  423101; NA; 1 in 400; Berg J, et al. 2013. J Exp Med. 210:2803.
 CD3 ; APC; biolegend;  300301; HIT3A; 1 in 80; Kaushal A, et al. 2021. Blood Cancer Discov. 2:600
 CD19; AF700; biolegend;  302225; HIB19; 1 in 80; Viny AD, et al. 2019. Cell Stem Cell. 25:682
 CD45RA ; PerCPCy5.5; biolegend;  304121; HI100; 1 in 80; Sammicheli S, et al. 2012. J Autoimmun. 38:304
 CCR7; BV711; biolegend;  353227; G043H7; 1 in 80; Arunachalam PS, et al. 2021. Nature. 596:410.
 CD14; BV605; biolegend;  367125; 63D3; 1 in 80; Perry JSA, et al. 2018. Immunity. 48:923.

Validation | These werre all previously used and validated antibodies.

## Plants

Seed stocks | NA

Novel plant genotypes | NA

Authentication | NA

