## [Peer Review File · Nature Genetics]

The long-term effects of chemotherapy on normal blood cells

Corresponding Author: Professor Michael Stratton

This manuscript has been previously reviewed at another journal. This document only contains information relating to versions considered at Nature Genetics.

Version 0:

Decision Letter:

30th Oct 2024

Dear Mike,

Your Article, "The long-term effects of chemotherapy on normal blood cells" has now been seen by the 3 original Nature referees. You will see from their comments below that while we are close to a final decision on publication, a few points still require consideration.

We remain interested in the possibility of publishing your study in Nature Genetics, but would like to consider your response to these concerns in the form of a revised manuscript before we make a final decision.

Briefly, two reviewers are satisfied, with no further comments and supporting publication; the other, however, has minor comments that may require further analysis and/or review.

To guide the scope of the revisions, the editors discuss the referee reports in detail within the team, including with the chief editor, with a view to identifying key priorities that should be addressed in revision and sometimes overruling referee requests that are deemed beyond the scope of the current study. We hope that you will find the prioritized set of referee points to be useful when revising your study. Please do not hesitate to get in touch if you would like to discuss these issues further.

We therefore invite you to revise your manuscript taking into account all reviewer and editor comments. Please highlight all changes in the manuscript text file. At this stage we will need you to upload a copy of the manuscript in MS Word .docx or similar editable format.

*2) If you have not done so already please begin to revise your manuscript so that it conforms to our Article format instructions, available

[here](http://www.nature.com/ng/authors/article_types/index.html).

*3) Include a revised version of any required Reporting Summary: <https://www.nature.com/documents/hr-reporting-summary.pdf>

Please be aware of our [guidelines](https://www.nature.com/nature-research/editorial-policies/image-integrity) on digital image standards.

Link Redacted

We hope to receive your revised manuscript within four to eight weeks. If you cannot send it within this time, please let us know.

Sincerely,

Michael Fletcher, PhD
Senior Editor, Nature Genetics
ORCID: 0000-0003-1589-7087

Reviewers' Comments:

Reviewer #1 (Remarks to the Author):

The reviewers have done an outstanding job in replying to the reviewer comments. I have no further comments.

Reviewer #2 (Remarks to the Author):

In this revised study, the authors characterize the clonal architecture and mutation burden in 23 individuals that had been exposed to cytotoxic therapy by performing whole genome sequencing of single cell-derived hematopoietic colonies. Better annotation of aging-associated single nucleotide variants and a more explicit description of study limitations have significantly improved the rigor of this study. Major findings remain the same and include the following:

- 1) Total mutation burden is increased in most cases with chemotherapy exposure and the mutational signature correlates with chemotherapy type. This is perhaps the weakest part of the study. Most of the correlations between specific chemotherapy and mutational signatures have previously been identified. Moreover, some of the associations identified in this study are based on single patients. These limitations are now outlined in the text.
- 2) Chemotherapy-induced mutations differ based on cell type. An interesting observation, although again based on limited data. Whether a generalizable finding is uncertain.
- 3) All of the single cell-derived HPSC colonies show a similar mutation burden/signature, suggesting that most HSPCs are active and susceptible to chemotherapy-induced changes. Another interesting and novel observation. I do have some concerns about the potential bias introduced by sequencing HSPCs able to form colonies. Again, this concern is now addressed in the text.
- 4) Chemotherapy exposure is associated with expansion of certain clones, often carrying mutations in DNA damage response genes, resulting in less clonal diversity. This is a well-established phenotype, reducing its impact. That said, the degree to which chemotherapy selects for DDR genes is striking and unexpected and has implications for mechanisms of clonal evolution.

Specific Comments

(1) This manuscript is significantly improved with the annotation of the aging-associated SBS1, SBS5, and SBS Blood mutations in Supplementary Table 9. For improved clarity, the authors should consider an extended figure dividing the SBS1, 5 and Blood mutations (clearly aging-associated) and the other mutations (which are not). This will help more accurately assess the 23 chemotherapy-associated cases, where there appears to be 4 cases with a very large non aging-associated mutation burden (>1324), 6 with a moderate burden (> 324), and 13 with no significant increased burden (< 34),

as opposed to 4, 13, and 6 respectively.

(2) In Supplemental Table 1, PD47703 and PD47700 are mislabeled (and the word "peripheral" is misspelled).

(3) Regarding oxaliplatin, the authors state that SBSF was present in a subset of oxaliplatin cases at low burdens. In Supplementary Table 9, of 8 individuals receiving oxaliplatin, only PD50307 had a burden > 10 (30) with 3 having a burden between 1-10. How confident can one be with a mutational signature having such a low burden (particularly those cases with < 10 mutations attributed to the signature)?

(4) Regarding PPM1D mutant clones and cytopenias, the authors speculate that their presence may play a role in reducing the regenerative ability of bone marrow potentially leading to cytopenias and infections. It's equally likely that the expansion of such clones is a marker of previous genotoxic stress that has suppressed their wild-type counterparts, resulting in poorer functioning HSCs. In mouse models, there is no evidence the presence of PPM1D mutant clones in the absence of cytotoxic stress promotes cytopenias.

(5) The authors state that in Figure 3 (MISSING FIGURE), the mutation burdens attributable to the platinum agents, procarbazine, and the nitrogen mustards were across all sampled HSPCs, suggesting few HSCs were protected from DNA damage. This is an important point and should be assessed statistically versus by gestalt. For example, in PD47703 (Extended Figure 5; Rebuttal Figure 16), certain HSCs have a significantly higher number of SBSF mutations than others, so presumably the cells without SBSF mutations were protected? In a similar vein, Rebuttal Figures 15 and 16 are very interesting data, and the authors may consider adding these to the manuscript as the effect of cytotoxic therapy (# mutations acquired and mutation distribution; SNVs vs. indels) on HSCs with mutations in DDR genes (TP53, PPM1D, CHEK2, etc.) versus WT HSCs would be of significant interest.

Reviewer #3 (Remarks to the Author):

The response to reviewers was outstanding.

All of my questions were answered suitably, and a significant amount of work was done to further clarify the manuscript and satisfy any objections the other reviewers and I raised. I also learned from these responses and am thankful to the authors for the well articulated, thoughtful rebuttal.

Version 1:

Decision Letter:

Our ref: NG-A66041R

9th Jan 2025

Dear Mike,

Happy New Year! Here's to a safe and prosperous 2025.

Thank you for submitting your revised manuscript "The long-term effects of chemotherapy on normal blood cells" (NG-A66041R). It has now been seen by the original referees and their comments are below. The reviewers find that the paper has improved in revision, and therefore we'll be happy in principle to publish it in Nature Genetics, pending minor revisions to satisfy the referees' final requests and to comply with our editorial and formatting guidelines.

As the current version of your manuscript is in a PDF format, please email us a copy of the file in an editable format (Microsoft Word or LaTeX)-- we can not proceed with PDFs at this stage.

Sincerely,

Michael Fletcher, PhD
Senior Editor, Nature Genetics
ORCID: 0000-0003-1589-7087

Reviewer #2 (Remarks to the Author):

The authors have adequately addressed my concerns.

6th December 2024

To the Editor,

Thank you for the remaining Referee comments to our revised manuscript "The long-term effects of chemotherapy on normal blood cells". Please find below our responses. As before, Referee comments are in black, our responses in blue, and changes we have made in red text. Text included from the main manuscript is highlighted with changes made in red text.

Reviewers' Comments:

Reviewer #1 (Remarks to the Author):

The reviewers have done an outstanding job in replying to the reviewer comments. I have no further comments.

We thank the Reviewer for these kind comments and time reviewing our study.

Reviewer #2 (Remarks to the Author):

In this revised study, the authors characterize the clonal architecture and mutation burden in 23 individuals that had been exposed to cytotoxic therapy by performing whole genome sequencing of single cell-derived hematopoietic colonies. Better annotation of aging-associated single nucleotide variants and a more explicit description of study limitations have significantly improved the rigor of this study. Major findings remain the same and include the following:

1) Total mutation burden is increased in most cases with chemotherapy exposure and the mutational signature correlates with chemotherapy type. This is perhaps the weakest part of the study. Most of the correlations between specific chemotherapy and mutational signatures have previously been identified. Moreover, some of the associations identified in this study are based on single patients. These limitations are now outlined in the text.

2) Chemotherapy-induced mutations differ based on cell type. An interesting observation, although again based on limited data. Whether a generalizable finding is uncertain.

3) All of the single cell-derived HPSC colonies show a similar mutation burden/signature, suggesting that most HSPCs are active and susceptible to chemotherapy-induced changes. Another interesting and novel observation. I do have some concerns about the potential bias introduced by sequencing HSPCs able to form colonies. Again, this concern is now addressed in the text.

4) Chemotherapy exposure is associated with expansion of certain clones, often carrying mutations in DNA damage response genes, resulting in less clonal diversity. This is a well-established phenotype, reducing its impact. That said, the degree to which chemotherapy selects for DDR genes is striking and unexpected and has implications for mechanisms of clonal evolution.

Specific Comments

(1) This manuscript is significantly improved with the annotation of the aging-associated SBS1, SBS5, and SBS Blood mutations in Supplementary Table 9. For improved clarity, the authors should consider an extended figure dividing the SBS1, 5 and Blood mutations (clearly aging-associated) and the other mutations (which are not). This will help more accurately assess the 23 chemotherapy-associated cases, where there appears to be 4 cases with a very large non aging-associated mutation burden (>1324), 6

with a moderate burden (> 324), and 13 with no significant increased burden (< 34), as opposed to 4, 13, and 6 respectively.

Thank you for this suggestion. We have now added Extended Data Fig.1 c-f splitting total mutation burden per donor into those mutations that are ageing-associated and those that are chemotherapy-associated (Rebuttal Figure 1). We plot high mutation and low mutation burden individuals separately for improved readability of plots. These plots more easily convey which individuals accumulated high levels of chemotherapy induced mutations, and that age associated mutations continue to accumulate in patients exposed to chemotherapy. As mentioned in the previous Rebuttal, for some individuals there is a slight underestimate in the assignment of age-associated mutations for some individuals when using *hdp*. However, this unbiased approach allows maximal extraction of de novo signatures. We have previously demonstrated that the expected number of age-associated mutations are present when we use other signature assignment tools with inputs of age-associated signatures as priors. Age associated mutations also appear to be underestimated in the figure below in some chemotherapy-exposed and normal individuals who have fewer unique age-associated mutations than expected due to the presence of clonally expanded clades.

Rebuttal Figure 1. (updated Extended Data Fig. 1) Mean sequencing depth in the normal and chemotherapy exposed HSPC colonies used for mutation burden analysis and patterns of mutation accumulation. a, Box plot representing the quartile distribution of mean sequencing depth in 90 colonies from normal (blue) and 189 colonies from chemotherapy exposed (red) individuals. b, Boxplot comparing the mean sequencing depth between Alkylating/ Platinum agent exposed and non-exposed colonies. The number of colonies in each agent group are shown at the top of the plot. c-f, HSPC single base substitutions associated with chemotherapy (c,d) and

age (e,f). CC, colorectal carcinoma; LC, lung cancer; NB, neuroblastoma; FL, follicular lymphoma; DLBL, diffuse large B cell lymphoma; MZL, marginal zone lymphoma; LL, lymphoplasmacytic lymphoma; M, multiple myeloma; HL, Hodgkin lymphoma; AML, acute myeloid leukaemia; 1, Platinum agents; 2, Alkylating agents; 3, Antimetabolites; 4, Topo I inhibitors; 5, Topo II inhibitors; 6, Vinca alkaloids; 7, Cytotoxic antibiotics.

(2) In Supplemental Table 1, PD47703 and PD47700 are mislabeled (and the word “peripheral” is misspelled).

Thank you for spotting these typos and our apologies for the oversight. We have now corrected these in the new file Supplementary_tables_20241031.xlsx. We have also checked over remaining material more thoroughly and have corrected typos in spelling on Extended Figure 5.

(3) Regarding oxaliplatin, the authors state that SBSF was present in a subset of oxaliplatin cases at low burdens. In Supplementary Table 9, of 8 individuals receiving oxaliplatin, only PD50307 had a burden > 10 (30) with 3 having a burden between 1-10. How confident can one be with a mutational signature having such a low burden (particularly those cases with < 10 mutations attributed to the signature)?

Thank you for raising this. We fully agree that we would not wish to make definitive conclusions regarding the presence of an oxaliplatin signature in HSPCs based on these data. However, there were several lines of evidence that suggested that these mutations may genuinely be related to oxaliplatin.

1. There were 12 samples across 7 donors with attributions of mutations to SBSF in our study (Supplementary Table 9). The two patients exposed to carboplatin/cisplatin had the highest SBSF (292-1711 mutations depending on cell type, 5 samples). Beyond this, there were fewer SBSF mutations (3 to 54) in the remaining 7 samples (from 5 donors). Of these 5 donors, 4 were exposed to oxaliplatin, save for the sample showing the fewest mutations (3 mutations assigned to SBSF in PD47538). This enrichment for low burden mutagenesis via SBSF in patients exposed to oxaliplatin gives us some confidence that this association is genuine.

2. In two donors exposed to oxaliplatin, SBSF was observed in two different cell types (PD47700 - B cells with 54 mutations and HSPCs with 30 mutations; PD47537 - Monocytes with 31 mutations and HSPCs with 7 mutations).

3. The trinucleotide context of SBSF observed for oxaliplatin is similar to that reported previously^{1,2}.

4. The observed lower mutation burden in HSCs due to oxaliplatin has been noted for other tissues. For example, the liver also appears relatively protected from oxaliplatin-induced mutagenesis², and future work would be required to understand the basis of these interesting tissue-specific protective mechanisms.

Balancing the uncertainty around mutation assignment due to the small numbers attributed against the observations above that suggest a genuine association, we have opted to state the following (page 11): “SBSF was found in individuals treated with carboplatin or cisplatin, and in a subset of oxaliplatin-treated individuals in whom it was present at much lower burdens.”

(4) Regarding PPM1D mutant clones and cytopenias, the authors speculate that their presence may play a role in reducing the regenerative ability of bone marrow potentially leading to cytopenias and infections. It's equally likely that the expansion of such clones is a marker of previous genotoxic stress that has suppressed their wild-type counterparts, resulting in poorer functioning HSCs. In mouse models, there is no evidence the presence of PPM1D mutant clones in the absence of cytotoxic stress promotes cytopenias.

We agree with the Reviewer regarding this alternative explanation. We have now adjusted the statement in the paper on page 18 as follows, “The changes in clonal architecture resulting from chemotherapy exposure have potential significance for two reasons: first, PPM1D mutant clones may themselves reduce the regenerative ability of the bone marrow^{3,4}, or the presence of PPM1D clones may simply be a marker of a more general state of reduced HSC function post chemotherapy. One may speculate that the presence of many such clones played a role in the cytopenias and infection in PD47703 post autograft. Secondly, selection of TP53 mutant clones confers a high risk of developing secondary myeloid malignancies, including AML as seen in PD37580, whose disease was treatment refractory and carried bi-allelic TP53 mutations in association with a complex karyotype.”

(5) The authors state that in Figure 3 (MISSING FIGURE), the mutation burdens attributable to the platinum agents, procarbazine, and the nitrogen mustards were across all sampled HSPCs, suggesting few HSCs were protected from DNA damage. This is an important point and should be assessed statistically versus by gestalt. For example, in PD47703 (Extended Figure 5; Rebuttal Figure 16), certain HSCs have a significantly higher number of SBS mutations than others, so presumably the cells without SBS mutations were protected? In a similar vein, Rebuttal Figures 15 and 16 are very interesting data, and the authors may consider adding these to the manuscript as the effect of cytotoxic therapy (# mutations acquired and mutation distribution; SNVs vs. indels) on HSCs with mutations in DDR genes (TP53, PPM1D, CHEK2, etc.) versus WT HSCs would be of significant interest.

First, we apologise for the omission of Figure 3, which we now realise was absent from the .pdf file. Please find it below. It is also reinserted back in the main manuscript file.

Figure 3| Phylogenetic trees and mutational signatures across a range of normal and chemotherapy exposed individuals. Phylogenies were constructed using shared mutation data and the algorithm MPBoot (Methods). Branch lengths correspond to SBS burdens. A stacked bar plot represents the signatures contributing to each branch with color code below the trees. SBSUnassigned indicates mutations that could not confidently be assigned to any reported signature. Drugs in parentheses are those received by the individual at the same time but not believed to be the mutagenic agent.

We accept that our original wording that ‘few HSCs were protected from DNA damage’, was rather generic. We have now adjusted this statement to (page 12, para 2) to: **Although there was some variability in the mutation burdens attributable to cisplatin/carboplatin, procarbazine, chlorambucil and**

bendamustine (the most highly mutagenic agents) across HSPCs from each individual (Fig. 3), the evidence suggests that there were no HSPCs completely protected from DNA damage.

As suggested by the Referee, we have now looked into heterogeneity of mutagenesis statistically. We consider a null hypothesis whereby all sampled HSCs of a donor have the same underlying exposures equal to the donor level average SBS96 proportions. The null distribution is then formed by multinomial resampling of each sample's SBS96 counts, performing signature attribution (`MutationalSignatures::fit_to_signature`), and then measuring the within-donor variance of the signature of interest. This process indicates that for larger trees the observed variance of the main chemotherapy signature attributions is significantly greater than expected under this null distribution (*Rebuttal Table 1*).

Donor	Signature	P
PD37580	SBSC	0.004
PD47541	SBSC	0.027
PD47541	SBSD	0.105
PD47699	SBSD	0.342
PD47699	SBSE	0.279
PD47703	SBSA	0.001
PD47703	SBSB	0.001
PD47703	SBSC	0.001
PD47703	SBSD	0.001
PD50306	SBSF	0.001
PD50307	SBSF	0.001
PD50308	SBSA	0.001
PD50308	SBSB	0.001

Rebuttal Table 1. Exposure to signatures is significantly heterogenous within donor. The table shows one-sided heterogeneity p-values indicating that many signatures exhibit greater variability than is consistent with the same underlying donor level exposure for each HSC.

The question then remains as to whether this heterogeneity is sufficient to “protect” some cells from the chemotherapy signatures. We consider a cell protected from a signature(s) if the fraction of cells with exposure greater than some threshold (say 1%) is less than expected by chance based on the above null distribution. We find that for the individuals with extensive chemotherapy exposure, no cell is spared damage at the 1% threshold in either observed exposures, or indeed in the Null. This threshold must be raised above 10% for us to observe any protected cells (see PD47699) and even then, there is no significant difference between the observed and the Null (*Rebuttal Figure 2*).

Rebuttal Figure 2 No HSCs are fully protected from composite chemotherapy induced mutagenesis. The plot shows the proportion of single derived colonies (y-axis) that exhibit more than a threshold level (“Protected Threshold” – x-axis) of the combined chemotherapy signatures. The violin plots show the distribution of both the observed proportions (blue) under resampling of representative samples for mutant clades and the corresponding multinomial resampling based null distribution (red).

Within an individual patient, we also looked further into whether some HSCs experienced more or less mutagenesis from one signature versus another. Whilst we do observe some evidence of protection at the level of individual signatures statistically (*Rebuttal Figure 3*), it is possible that the high level of similarity between some of the chemotherapy signatures (e.g. SBSC and SBSD; cosine similarity 0.87) make the attribution of individual signatures unstable. The impact is less evident when assessed at the level of combined attribution (*Rebuttal Figure 2 (above) and 3 (below, right panel)*) which is much more stable.

Rebuttal Figure 3. Some cells of PD47703 appear “protected” from specific signatures. A subset of HSCs appear significantly protected from SBSC ($P < 0.001$) for thresholds $\leq 5\%$. Much fewer are protected from the composite effect of SBSC+SBSD.

In our previous Rebuttal Figure 16, we showed that there appeared to be enrichment for SBSD within branches of expanded clades from PD47703 (Rebuttal Figure 4).

Rebuttal Figure 4. MutationalPatterns reattributed per branch signatures for PD47703.

We considered whether this finding could have been technically affected by the branching pattern of the tree. Therefore, we assessed whether this was still the case when the attribution is performed at the level of individual unrelated samples rather than at the level of branches.

Rebuttal Figure 5: Clade vs Unrelated signature proportions for PD47703. Expanded clades here are defined as expansions with a most recent common ancestor occurring after 100 units molecular time. To

ensure independent sampling the clades are each represented by a single sample averaged over the clade. Note that similar results, although not always significant, are found if instead random individual HSCs are chosen to represent the expanded clades.

The p-values above not adjusted for multiple testing. The analysis suggests that even at the single sample level, SBSB and SBSA are present in higher proportions in samples that are in a clonal expansion (mean= 0.14) vs singleton samples (mean=0.08). However, it is clear that we are insufficiently powered to robustly draw this conclusion. It is also clearly the case that patterns of signature assignments in PD47703 are somewhat affected by SBSB being more effectively assigned in branches after late coalescences. Overall, we do not wish to draw definite conclusions from this single phylogenetic tree. Of note, we have opted for making the Review files public and therefore, readers would be able to read these further discussions if such observations are of interest.

Reviewer #3 (Remarks to the Author):

The response to reviewers was outstanding. All of my questions were answered suitably, and a significant amount of work was done to further clarify the manuscript and satisfy any objections the other reviewers and I raised. I also learned from these responses and am thankful to the authors for the well articulated, thoughtful rebuttal.

We're delighted with this feedback and we wish to thank the Reviewer for their time and comments that have helped significantly improve this manuscript.